# Energy-based Out-of-Distribution Detection for Graph Neural Networks

**Qitian Wu, Yiting Chen, Chenxiao Yang, Junchi Yan**[*]
Department of CSE & MoE Lab of Artificial Intelligence, Shanghai Jiao Tong University
{echo740, sjtucyt, chr26195, yanjunchi}@sjtu.edu.cn

## Abstract

Learning on graphs, where instance nodes are inter-connected, has become one of the central problems for deep learning, as relational structures are pervasive and induce data inter-dependence which hinders trivial adaptation of existing approaches that assume inputs to be i.i.d. sampled. However, current models mostly focus on improving testing performance of in-distribution data and largely ignore the potential risk w.r.t. out-of-distribution (OOD) testing samples that may cause negative outcome if the prediction is overconfident on them. In this paper, we investigate the under-explored problem, OOD detection on graph-structured data, and identify a provably effective OOD discriminator based on an energy function directly extracted from graph neural networks trained with standard classification loss. This paves a way for a simple, powerful and efficient OOD detection model for GNN-based learning on graphs, which we call GNNSAFE. It also has nice theoretical properties that guarantee an overall distinguishable margin between the detection scores for in-distribution and OOD samples, which, more critically, can be further strengthened by a learning-free energy belief propagation scheme. For comprehensive evaluation, we introduce new benchmark settings that evaluate the model for detecting OOD data from both synthetic and real distribution shifts (cross-domain graph shifts and temporal graph shifts). The results show that GNNSAFE achieves up to $17.0\%$ AUROC improvement over state-of-the-arts and it could serve as simple yet strong baselines in such an under-developed area. The codes are available at https://github.com/qitianwu/GraphOOD-GNNSafe.

## 1 Introduction

Real-world applications often require machine learning systems to interact with an open world, violating the common assumption that testing and training distributions are identical. This urges the community to devote increasing efforts on how to enhance models' *generalization* (Muandet et al., 2013) and *reliability* (Liang et al., 2018) w.r.t. out-of-distribution (OOD) data. However, most of current approaches are built on the hypothesis that data samples are independently generated (e.g., image recognition where instances have no interaction). Such a premise hinders these models from readily adapting to graph-structured data where node instances have inter-dependence (Zhao et al., 2020; Ma et al., 2021; Wu et al., 2022a).

**Out-of-Distribution Generalization.** To fill the research gap, a growing number of recent studies on graph-related tasks move beyond the single target w.r.t. in-distribution testing performance and turn more attentions to how the model can *generalize* to perform well on OOD data. One of the seminal works (Wu et al., 2022a) formulates the graph-based OOD generalization problem and leverages (causal) invariance principle for devising a new domain-invariant learning approach for graph data. Different from grid-structured and independently generated images, distribution shifts concerning graph-structured data can be more complicated and hard to address, which often requires graph-specific technical originality. For instances, Yang et al. (2022c) proposes to identify invariant substructures, i.e., a subset of nodes with causal effects to labels, in input graphs to learn

---

[*]Corresponding author: Junchi Yan who is also affiliated with Shanghai AI Lab. The work was in part supported by National Key Research and Development Program of China (2020AAA0107600), National Natural Science Foundation of China (62222607), STCSM (22511105100).

stable predictive relations across environments, while Yang et al. (2022b) resorts to an analogy of thermodynamics diffusion on graphs to build a principled knowledge distillation model for geometric knowledge transfer and generalization.

**Out-of-Distribution Detection.** One critical challenge that stands in the way for trustworthy AI systems is how to equip the deep learning models with enough *reliability* of realizing *what they don't know*, i.e., detecting OOD data on which the models are expected to have low confidence (Amodei et al., 2016; Liang et al., 2018). This is fairly important when it comes to safety-critical applications such as medical diagnosis (Kukar, 2003), autonomous driving (Dai & Van Gool, 2018), etc. Such a problem is called out-of-distribution detection in the literature, which aims at discriminating OOD data from the in-distribution one. While there are a surge of recent works exploring various effective methods for OOD detection (Hendrycks & Gimpel, 2016; Bevandić et al., 2018; DeVries & Taylor, 2018; Hein et al., 2019; Hsu et al., 2020; Sun et al., 2021; Bitterwolf et al., 2022), these models again tacitly assume inputs to be independently sampled and are hard to be applied for graph data.

In this paper, we investigate into out-of-distribution detection for learning on graphs, where the model needs to particularly handle data inter-dependence induced by the graph. Our methodology is built on graph neural networks (GNNs) as an encoder for node representation/prediction that accommodates the structural information. As an important step to enhance the reliability of GNN models against OOD testing instances, we identify an intrinsic OOD discriminator from a GNN classifier trained with standard learning objective. *The key insight of our work is that standard GNN classifiers possess inherent good capability for detecting OOD samples (i.e., what they don't know) from unknown testing observations*, with details of the model as described below.

○ *Simplicity and Generality:* The out-of-distribution discriminator in our model is based on an energy function that is directly extracted through simple transformation from the predicted logits of a GNN classifier trained with standard supervised classification loss on in-distribution data. Therefore, our model can be efficiently deployed in practice, i.e., it does not require training a graph generative model for density estimation or any extra OOD discriminator. Also, the model keeps a general form, i.e., the energy-based detector is agnostic to GNNs' architectures and can in principle enhance the reliability for arbitrary off-the-shelf GNNs (or more broadly, graph Transformers) against OOD data.

○ *Theoretical Soundness:* Despite simplicity, our model can be provably effective for yielding distinguishable scores for in-distribution and OOD inputs, which can be further reinforced by an energy-based belief propagation scheme, a learning-free approach for boosting the detection consensus over graph topology. We also discuss how to properly incorporate an auxiliary regularization term when training data contains additional OOD observation as outlier exposure, with double guarantees for preserving in-distribution learning and enhancing out-of-distribution reliability.

○ *Practical Efficacy:* We apply our model to extensive node classification datasets of different properties and consider various OOD types. When training with standard cross-entropy loss on pure in-distribution data and testing on OOD detection at inference time, our model consistently outperforms SOTA approaches with an improvement of up to $12.9\%$ on average AUROC; when training with auxiliary OOD exposure data as regularization and testing on new unseen OOD data, our model outperforms the strong competitors with up to $17.0\%$ improvement on average AUROC.

## 2 BACKGROUND

**Predictive tasks on graphs.** We consider a set of instances with indices $i \in \{1, 2, \cdots, N\} = \mathcal{I}$ whose generation process involves inter-dependence among each other, represented by an observed graph $G = (V, E)$ where $V = \{i | 1 \leq i \leq N\}$ denotes the node set containing all the instances and $E = \{e_{ij}\}$ denotes the edge set. The observed edges induce an adjacency matrix $A = [a_{ij}]_{N \times N}$ where $a_{ij} = 1$ if there exists an edge connecting nodes $i$ and $j$ and 0 otherwise. Moreover, each instance $i$ has an input feature vector denoted by $\mathbf{x}_i \in \mathbb{R}^D$ and a label $y_i \in \{1, \cdots, C\}$ where $D$ is the input dimension and $C$ denotes the class number. The $N$ instances are partially labeled and we define $\mathcal{I}_s$ (resp. $\mathcal{I}_u$) as the labeled (resp. unlabeled) node set, i.e., $\mathcal{I} = \mathcal{I}_s \cup \mathcal{I}_u$. The goal of standard (semi-)supervised learning on graphs is to train a node-level classifier $f$ with $\hat{Y} = f(X, A)$, where $X = [\mathbf{x}_i]_{i \in \mathcal{I}}$ and $\hat{Y} = [\hat{\mathbf{y}}_i]_{i \in \mathcal{I}}$, that predicts the labels for in-distribution instances in $\mathcal{I}_u$.

**Out-of-distribution detection.** Besides decent predictive performance on in-distribution testing nodes (sampled from the same distribution as training data), we expect the learned classifier is capable

for detecting out-of-distribution (OOD) instances that has distinct data-generating distribution with the in-distribution ones. More specifically, the goal of OOD detection is to find a proper decision function $G$ (that is associated with the classifier $f$) such that for any given input $\mathbf{x}$:

$$G(\mathbf{x}, \mathcal{G}_{\mathbf{x}}; f) = \begin{cases} 1, & \mathbf{x} \text{ is an in-distribution instance,} \\ 0, & \mathbf{x} \text{ is an out-of-distribution instance,} \end{cases} \tag{1}$$

where $\mathcal{G}_{\mathbf{x}} = (\{\mathbf{x}_j\}_{j \in \mathcal{N}_{\mathbf{x}}}, A_{\mathbf{x}})$ denotes the ego-graph centered at node $\mathbf{x}$, $\mathcal{N}_{\mathbf{x}}$ is the set of nodes in the ego-graph (within a certain number of hops) of $\mathbf{x}$ on the graph and $A_{\mathbf{x}}$ is its associated adjacency matrix. Our problem formulation here considers neighbored nodes that have inter-dependence with the centered node throughout the data-generating process for determining whether it is OOD. In such a case, the OOD decision for each input is dependent on other nodes in the graph, which is fairly different from existing works in vision that assume i.i.d. inputs and independent decision for each.

**Energy-based model.** Recent works (Xie et al., 2016; Grathwohl et al., 2020) establish an equivalence between a neural classifier and an energy-based (EBM) (Ranzato et al., 2007) when the inputs are i.i.d. generated. Assume a classifier as $h(\mathbf{x}) : \mathbb{R}^D \to \mathbb{R}^C$ which maps an input instance to a $C$-dimensional logits, and a softmax function over the logits gives a predictive categorical distribution: $p(y|\mathbf{x}) = \frac{e^{h(\mathbf{x})[y]}}{\sum_{c=1}^{C} e^{h(\mathbf{x})[c]}}$, where $h(\mathbf{x})_{[c]}$ queries the $c$-th entry of the output. The energy-based model builds a function $E(\mathbf{x}, y) : \mathbb{R}^D \to \mathbb{R}$ that maps each input instance with an arbitrarily given class label to a scalar value called *energy*, and induces a Boltzmann distribution: $p(y|\mathbf{x}) = \frac{e^{-E(\mathbf{x},y)}}{\sum_{y'} e^{-E(\mathbf{x},y')}} = \frac{e^{-E(\mathbf{x},y)}}{e^{-E(\mathbf{x})}}$, where the denominator $\sum_{y'} e^{-E(\mathbf{x},y')}$ is often called the partition function which marginalizes over $y$. The equivalance between an EBM and a discriminative neural classifier is established by setting the energy as the predicted logit value $E(\mathbf{x}, y) = -h(\mathbf{x})_{[y]}$. And, the energy function $E(\mathbf{x})$ for any given input can be

$$E(\mathbf{x}) = -\log \sum_{y'} e^{-E(\mathbf{x},y')}. \tag{2}$$

The energy score $E(\mathbf{x})$ can be an effective indicator for OOD detection due to its nice property: the yielded energy scores for in-distribution data overall incline to be lower than those of OOD data (Liu et al., 2020). However, to the best of our knowledge, current works on energy-based modeling as well as its application for OOD detection are mostly focused on i.i.d. inputs and the power of the energy model for modeling inter-dependent data has remained under-explored.

## 3 PROPOSED METHOD

We describe our new model for out-of-distribution detection in (semi-)supervised node classification where instances are inter-dependent. We will first present a basic version of our model with energy-based modeling (Section 3.1) and a novel energy-based belief propagation scheme (Section 3.2). After that, we extend the model with auxiliary OOD training data (if available) (Section 3.3).

### 3.1 ENERGY-BASED OOD DETECTION WITH DATA DEPENDENCE

To capture dependence among data points, the most commonly adopted model class follows the classic idea of Graph Neural Networks (GNNs) (Scarselli et al., 2008) which iteratively aggregate neighbored nodes' features for updating centered nodes' representations. Such an operation is often called *message passing* that plays a central role in modern GNNs for encoding topological patterns and accommodating global information from other instances. In specific, assume $\mathbf{z}_i^{(l)}$ as the representation of instance node $i$ at the $l$-th layer and the Graph Convolutional Network (GCN) (Kipf & Welling, 2017) updates node representation through a layer-wise normalized feature propagation:

$$Z^{(l)} = \sigma\left(D^{-1/2}\tilde{A}D^{-1/2}Z^{(l-1)}W^{(l)}\right), \quad Z^{(l-1)} = [\mathbf{z}_i^{(l-1)}]_{i \in \mathcal{I}}, \quad Z^{(0)} = X, \tag{3}$$

where $\tilde{A}$ is an adjacency matrix with self-loop on the basis of $A$, $D$ is its associated diagonal degree matrix, $W^{(l)}$ is the weight matrix at the $l$-th layer and $\sigma$ is a non-linear activation.

With $L$ layers of graph convolution, the GCN model outputs a $C$-dimensional vector $\mathbf{z}_i^{(L)}$ as logits for each node, denoted as $h_\theta(\mathbf{x}_i, \mathcal{G}_{\mathbf{x}_i}) = \mathbf{z}_i^{(L)}$, where $\theta$ denotes the trainable parameters of the GNN

model $h$. Induced by the GNN's updating rule, here the predicted logits for instance $\mathbf{x}$ are denoted as a function of the $L$-order ego-graph centered at instance $\mathbf{x}$, denoted as $\mathcal{G}_{\mathbf{x}}$. The logits are used to derive a categorical distribution with the softmax function for classification:

$$p(y \mid \mathbf{x}, \mathcal{G}_{\mathbf{x}}) = \frac{e^{h_\theta(\mathbf{x}, \mathcal{G}_{\mathbf{x}})_{[y]}}}{\sum_{c=1}^{C} e^{h_\theta(\mathbf{x}, \mathcal{G}_{\mathbf{x}})_{[c]}}}. \tag{4}$$

By connecting it to the EBM model that defines the relation between energy function and probability density, we can obtain the energy form induced by the GNN model, i.e., $E(\mathbf{x}, \mathcal{G}_{\mathbf{x}}, y; h_\theta) = -h_\theta(\mathbf{x}, \mathcal{G}_{\mathbf{x}})_{[y]}$. Notice that the energy presented above is directly obtained from the predicted logits of the GNN classifier (in particular, without changing the parameterization of GNNs). And, the free energy function $E(\mathbf{x}, \mathcal{G}_{\mathbf{x}}; h_\theta)$ that marginalizes $y$ can be expressed in terms of the denominator in Eqn. 4:

$$E(\mathbf{x}, \mathcal{G}_{\mathbf{x}}; h_\theta) = -\log \sum_{c=1}^{C} e^{h_\theta(\mathbf{x}, \mathcal{G}_{\mathbf{x}})_{[c]}}. \tag{5}$$

The non-probabilistic energy score given by Eqn. 5 accommodates the information of the instance itself and other instances that have potential dependence based on the observed structures.

For node classification, the GNN model is often trained by minimizing the negative log-likelihood of labeled training data (i.e., supervised classification loss)

$$\mathcal{L}_{sup} = \mathbb{E}_{(\mathbf{x}, \mathcal{G}_{\mathbf{x}}, y) \sim \mathcal{D}_{in}} \left( -\log p(y \mid \mathbf{x}, \mathcal{G}_{\mathbf{x}}) \right) = \sum_{i \in \mathcal{I}_s} \left( -h_\theta(\mathbf{x}_i, \mathcal{G}_{\mathbf{x}_i})_{[y_i]} + \log \sum_{c=1}^{C} e^{h_\theta(\mathbf{x}_i, \mathcal{G}_{\mathbf{x}_i})_{[c]}} \right), \tag{6}$$

where $\mathcal{D}_{in}$ denotes the distribution where the labeled portion of training data is sampled. We next show that, for any GNN model trained with Eqn. 6, the energy score can provably be a well-posed indicator of whether the input data is from in-distribution or out-of-distribution.

**Proposition 1.** The gradient descent on $\mathcal{L}_{sup}$ will overall decrease the energy $E(\mathbf{x}, \mathcal{G}_{\mathbf{x}}; h_\theta)$ for any in-distribution instance $(\mathbf{x}, \mathcal{G}_{\mathbf{x}}, y) \sim \mathcal{D}_{in}$.

The proposition shows that the induced energy function $E(\mathbf{x}, \mathcal{G}_{\mathbf{x}}; h_\theta)$ from the supervisedly trained GNN guarantees an overall trend that the energy scores for in-distribution data will be pushed down. This suggests that it is reasonable to use the energy scores for detecting OOD inputs at inference time.

**Remark.** The results above are agnostic to GNN architectures, i.e., the energy-based OOD discriminator can be applied for arbitrary off-the-shelf GNNs. Furthermore, beyond defining message passing on fixed input graphs, recently proposed graph Transformers Wu et al. (2022b; 2023) are proven to be powerful encoders for node-level prediction. The energy-based model is also applicable for these Transformer-like models and we leave such an extension as future works.

### 3.2 CONSENSUS BOOSTING WITH ENERGY-BASED BELIEF PROPAGATION

As the labeled data is often scarce in the semi-supervised learning setting, the energy scores that are directly obtained from the GNN model trained with supervised loss on labeled data may not suffice to generalize well. The crux for enhancing the generalization ability lies in how to leverage the unlabeled data in training set that could help to better recognize the geometric structures behind data (Belkin et al., 2006; Chong et al., 2020). Inspired by the *label propagation* (Zhu et al., 2003), a classic non-parametric semi-supervised learning algorithm, we propose energy-based belief propagation that iteratively propagates estimated energy scores among inter-connected nodes over the observed structures. We define $\mathbf{E}^{(0)} = [E(\mathbf{x}_i, \mathcal{G}_{\mathbf{x}_i}; h_\theta)]_{i \in \mathcal{I}}$ as a vector of initial energy scores for nodes in the graph and consider the propagation updating rule

$$\mathbf{E}^{(k)} = \alpha \mathbf{E}^{(k-1)} + (1-\alpha) D^{-1} A \mathbf{E}^{(k-1)}, \quad \mathbf{E}^{(k)} = [E_i^{(k)}]_{i \in \mathcal{I}}, \tag{7}$$

where $0 < \alpha < 1$ is a parameter controlling the concentration on the energy of the node itself and other linked nodes. After $K$-step propagation, we can use the final result for estimation, i.e., $\tilde{E}(\mathbf{x}_i, \mathcal{G}_{\mathbf{x}_i}; h_\theta) = E_i^{(K)}$, and the decision criterion for OOD detection can be

$$G(\mathbf{x}, \mathcal{G}_{\mathbf{x}}; h_\theta) = \begin{cases} 1, & \text{if} \quad \tilde{E}(\mathbf{x}, \mathcal{G}_{\mathbf{x}}; h_\theta) \leq \tau, \\ 0, & \text{if} \quad \tilde{E}(\mathbf{x}, \mathcal{G}_{\mathbf{x}}; h_\theta) > \tau, \end{cases} \tag{8}$$

where $\tau$ is the threshold and the return of 0 indicates the declaration for OOD. The rationale behind the design of energy belief propagation is to accommodate the physical mechanism in data generation with instance-wise interactions. Since the input graph could reflect the geometric structures among data samples with certain proximity on the manifold, it is natural that the generation of a node is conditioned on its neighbors, and thereby connected nodes tend to be sampled from similar distributions[1]. Given that the energy score of Eqn. 5 is an effective indicator for out-of-distribution, the propagation in the energy space essentially imitates such a physical mechanism of data generation, which can presumably reinforce the confidence on detection. Proposition 2 formalizes the overall trend of enforcing consensus for instance-level OOD decisions over graph topology.

**Proposition 2.** For any given instance $\mathbf{x}_i$, if the averaged energy scores of its one-hop neighbored nodes, i.e., $\frac{\sum_j a_{ij} E_j^{(k-1)}}{\sum_j a_{ij}}$, is less (resp. more) than the energy of its own, i.e., $E_i^{(k-1)}$, then the updated energy given by Eqn. 7 yields $E_i^{(k)} < E_i^{(k-1)}$ (resp. $E_i^{(k)} > E_i^{(k-1)}$).

Proposition 2 implies that the propagation scheme would push the energy towards the majority of neighbored nodes, which can help to amplify the energy gap between in-distribution samples and the OOD. This result is intriguing, suggesting that the energy model can be boosted by the simple propoagation scheme at inference time without any extra cost for training. We will verify its effectiveness through empirical comparison in experiments.

## 3.3 ENERGY-REGULARIZED LEARNING WITH BOUNDING GUARANTEES

The model presented so far does not assume any OOD data exposed to training. As an extension of our model, we proceed to consider another setting in prior art of OOD detection, e.g., (Hendrycks et al., 2019b; Liu et al., 2020; Bitterwolf et al., 2022), using extra OOD training data (often called OOD exposure). In such a case, we can add hard constraints on the energy gap through a regularization loss $\mathcal{L}_{reg}$ that bounds the energy for in-distribution data, i.e., labeled instances in $\mathcal{I}_s$, and OOD data, i.e., another set of auxiliary training instances $\mathcal{I}_o$ from a distinct distribution. And, we can define the final training objective as $\mathcal{L}_{sup} + \lambda \mathcal{L}_{reg}$, where $\lambda$ is a trading weight. There is much design flexibility for the regularization term $\mathcal{L}_{reg}$, and here we instantiate it as bounding constraints for absolute energy (Liu et al., 2020): $\mathcal{L}_{reg} =$

$$\frac{1}{|\mathcal{I}_s|} \sum_{i \in \mathcal{I}_s} \left( \text{ReLU} \left( \tilde{E} \left( \mathbf{x}_i, \mathcal{G}_{\mathbf{x}_i}; h_\theta \right) - t_{in} \right) \right)^2 + \frac{1}{|\mathcal{I}_o|} \sum_{j \in \mathcal{I}_o} \left( \text{ReLU} \left( t_{out} - \tilde{E} \left( \mathbf{x}_j, \mathcal{G}_{\mathbf{x}_j}; h_\theta \right) \right) \right)^2. \quad (9)$$

The above loss function will push the energy values within $[t_{in}, t_{out}]$ to be lower (resp. higher) for in-distribution (resp. out-of-distribution) samples, and this is what we expect.

The lingering concern, however, is that the introduction of an extra loss term may interfere with the original supervised learning objective $\mathcal{L}_{sup}$. In other words, the goal of energy-bounded regularization might be inconsistent with that of supervised learning, and the former may change the global optimum to somewhere the GNN model gives sub-optimal fitting on the labeled instances. Fortunately, the next proposition serves as a justification for the energy regularization in our model.

**Proposition 3.** For any energy regularization $\mathcal{L}_{reg}$, satisfying that the GNN model $h_{\theta^*}$ minimizing $\mathcal{L}_{reg}$ yields the energy scores upper (resp. lower) bounded by $t_{in}$ (resp. $t_{out}$) for in-distribution (resp. OOD) samples, where $t_{in} < t_{out}$ are two margin parameters, then the corresponding softmax categorical distribution $p(y|\mathbf{x}, \mathcal{G}_{\mathbf{x}}; h_{\theta^*})$ also minimizes $\mathcal{L}_{sup}$.

The implication of this proposition is important, suggesting that our used $\mathcal{L}_{reg}$ whose optimal energy scores are guaranteed for detecting OOD inputs and stay consistent with the supervised training of the GNN model. We thereby obtain a new training objective which strictly guarantees a desired discrimination between in-distribution and OOD instances in the training set, and in the meanwhile, does not impair model fitting on in-distribution data.

Admittedly, one disadvantage of the energy-regularized learning is that it requires additional OOD data exposed during training. As we will show in the experiments, nonetheless, the energy regularization turns out to be not a necessity in quite a few cases in practice.

---

[1]The connected nodes can have different labels or features, even if their data distributions are similar. Here the data distribution is a joint distribution regarding node feature, labels and graph structures.

## 4 EXPERIMENTS

The performance against out-of-distribution data on testing set can be measured by its capability of discriminating OOD testing samples from the in-distribution ones. In the meanwhile, we do not expect the degradation of the model's performance on in-distribution testing instances. An ideal model should desirably handle OOD detection without sacrificing the in-distribution accuracy.

### 4.1 SETUP

**Implementation details**   We call our model trained with pure supervised loss as GNNSAFE and the version trained with additional energy regularization as GNNSAFE++. We basically set the propagation layer number $K$ as 2 and weight $\alpha$ as 0.5. For fair comparison, the GCN model with layer depth 2 and hidden size 64 is used as the backbone encoder for all the model.

**Datasets and Splits**   Our experiments are based on five real-world datasets that are widely adopted as node classification benchmarks: `Cora`, `Amazon-Photo`, `Coauthor-CS`, `Twitch-Explicit` and `ogbn-Arxiv`. Following a recent work on OOD generalization over graphs (Wu et al., 2022a), we generally consider two ways for creating OOD data in experiments, which can reflect the real-world scenarios involving OOD instances inter-connected in graphs. 1) *Multi-graph scenario:* the OOD instances come from a different graph (or subgraph) that is dis-connected with any node in the training set. 2) *Single-graph scenario:* the OOD testing instances are within the same graph as the training instances, yet the OOD instances are unseen during training. Specifically,

○ For `Twitch`, we use the subgraph DE as in-distribution data and other five subgraphs as OOD data. The subgraph ENGB is used as OOD exposure for training of GNNSAFE++. These subgraphs have different sizes, edge densities and degree distributions, and can be treated as samples from different distributions Wu et al. (2022a).

○ For `Arxiv` that is a single graph, we partition the nodes into two-folds: the papers published before 2015 are used as in-distribution data, and the papers after 2017 are used as OOD data. The papers published at 2015 and 2016 are used as OOD exposure during training.

○ For `Cora`, `Amazon` and `Coauthor` that have no clear domain information, we synthetically create OOD data via three ways. *Structure manipulation:* use the original graph as in-distribution data and adopt stochastic block model to randomly generate a graph for OOD data. *Feature interpolation:* use random interpolation to create node features for OOD data and the original graph as in-distribution data. *Label leave-out:* use nodes with partial classes as in-distribution and leave out others for OOD.

For in-distribution data in each dataset, only partial nodes' labels are exposed for supervised training and another subset of nodes is used for validation. Other nodes are used as in-distribution testing data. We provide detailed dataset and splitting information in Appendix B.1.

**Metric**   We follow common practice using AUROC, AUPR and FPR95 as evaluation metrics for OOD detection. The in-distribution performance is measured by the Accuracy on testing nodes. In Appendix B.3 we provide more information about these evaluation metrics.

**Competitors**   We compare with two classes of models regarding OOD detection. The first family of baseline models focus on OOD detection in vision where inputs (e.g., images) are assumed to be i.i.d. sampled: *MSP* (Hendrycks & Gimpel, 2016), *ODIN* (Liang et al., 2018), *Mahalanobis* (Lee et al., 2018), *OE* (Hendrycks et al., 2019b) (using OOD exposure for training), *Energy* and *Energy Fine-Tune* (Liu et al., 2020) which also uses OOD exposure for training. We replace their convolutional neural network backbone with the GCN encoder used by our model for processing graph-structured data. The second family of baseline models are particularly designed for handling OOD data in graph machine learning and we compare with two SOTA approaches: the Graph-based Kernel Dirichlet GCN method *GKDE* (Zhao et al., 2020) and Graph Posterior Network *GPN* (Stadler et al., 2021). We provide more implementation details for these model in Appendix B.2.

### 4.2 COMPARATIVE RESULTS

**How does GNNSAFE++ perform compared with competitive methods?**   In Table 1 and 2 we report the experimental results of GNNSAFE++ in comparison with the competitive models. We found that GNNSAFE++ consistently outperform all competitors by a large margin throughout all

Table 1: Out-of-distribution detection results measured by **AUROC** (↑) / **AUPR** (↑) / **FPR95** (↓) on `Twitch`, where nodes in different sub-graphs are OOD data, and `Arxiv` dataset, where papers published after 2017 are OOD data. The in-distribution testing accuracy is reported for calibration. Detailed results on each OOD dataset (i.e., sub-graph or year) are presented in Appendix C. GPN reports out-of-memory issue on `Arxiv` with a 24GB GPU.

| Model | OOD Expo | Twitch | | | | Arxiv | | | |
|---|---|---|---|---|---|---|---|---|---|
| | | AUROC | AUPR | FPR | ID ACC | AUROC | AUPR | FPR | ID ACC |
| MSP | No | 33.59 | 49.14 | 97.45 | 68.72 | 63.91 | **75.85** | 90.59 | 53.78 |
| ODIN | No | **58.16** | **72.12** | 93.96 | 70.79 | 55.07 | 68.85 | 100.0 | 51.39 |
| Mahalanobis | No | 55.68 | 66.42 | **90.13** | 70.51 | 56.92 | 69.63 | 94.24 | 51.59 |
| Energy | No | 51.24 | 60.81 | 91.61 | 70.40 | **64.20** | 75.78 | 90.80 | 53.36 |
| GKDE | No | 46.48 | 62.11 | 95.62 | 67.44 | 58.32 | 72.62 | 93.84 | 50.76 |
| GPN | No | 51.73 | 66.36 | 95.51 | 68.09 | - | - | - | - |
| GNNSAFE | No | **66.82** | 70.97 | **76.24** | 70.40 | **71.06** | **80.44** | **87.01** | 53.39 |
| OE | Yes | 55.72 | 70.18 | 95.07 | 70.73 | 69.80 | 80.15 | 85.16 | 52.39 |
| Energy FT | Yes | **84.50** | **88.04** | **61.29** | 70.52 | **71.56** | **80.47** | **80.59** | 53.26 |
| GNNSAFE++ | Yes | **95.36** | **97.12** | **33.57** | 70.18 | **74.77** | **83.21** | **77.43** | 53.50 |

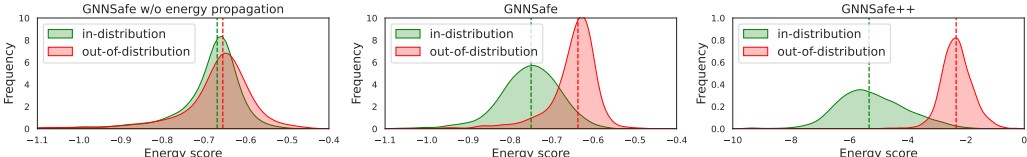

(a) The energy distributions on `Twitch` where nodes in different sub-graphs are OOD instances

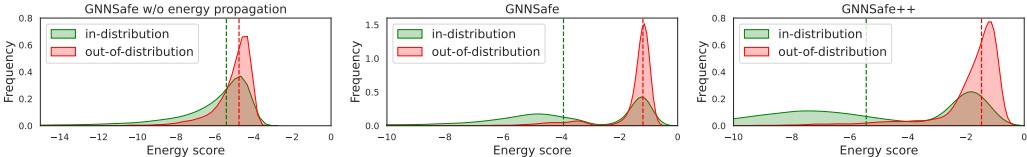

(b) The energy distributions on `Arxiv` where nodes appearing in the future are OOD instances

Figure 1: Comparison of the energy-based OOD detection performance yielded by vanilla energy model (described in Section 3.1), using energy-based propagation (proposed in Section 3.2) and additionally using energy-regularized training (proposed in Section 3.3). These ablation results demonstrate that the proposed energy propagation and regularized training can both be helpful for enlarging the energy margin between in-distribution and OOD inputs.

cases, increasing average AUROC by $12.8\%$ (resp. $17.0\%$) and reducing average FPR95 by $44.8\%$ (resp. $21.0\%$) on `Twitch` (resp. `Cora-Structure`) and maintain comparable in-distribution accuracy. This suggests the superiority of our energy-based model for OOD detection on graphs. Furthermore, without using the energy regularization, GNNSAFE still achieves very competitive OOD detection results, increasing average AUROC by $12.9\%$ on `Cora-Structure`. In particular, it even outperforms OE on all datasets and exceeds Energy FT in most cases on `Cora`, `Amazon` and `Coauthor`, though these two competitors use extra OOD exposure for training. This shows a key merit of our proposed model: it can enhance the reliability of GNNs against OOD data without costing extra computation resources. To quantitatively show the efficiency advantage, we compare the training and inference times in Table 8 in Appendix C. We can see that GNNSAFE is much faster than the SOTA model GPN during both training and inference and as efficient as other competitors. These results altogether demonstrate the promising efficacy of GNNSAFE++ for detecting OOD anomalies in real graph-related applications where the effectiveness and efficiency are both key considerations.

**How does the energy propagation improve inference-time OOD detection?** We study the effectiveness of the proposed energy-based belief propagation by comparing the performance of Energy and GNNSAFE (the Energy model in our setting can be seen as ablating the propagation module from GNNSAFE). Results in Table 1 and 2 consistently show that the propagation scheme indeed brings up significant performance improvement. Furthermore, in Fig. 1 we compare the energy scores produced by GNNSAFE (the middle column) and Energy (the left column). The figures show that the propagation scheme can indeed help to separate the scores for in-distribution and OOD inputs.

Table 2: Out-of-distribution detection performance measured by **AUROC** (↑) on datasets `Cora`, `Amazon`, `Coauthor` with three OOD types (**S**tructure manipulation, **F**eature interpolation, **L**abel leave-out). Other results for AUPR, FPR95 and in-distribution accuracy are deferred to Appendix C.

| **Model** | **OOD Expo** | Cora S | Cora F | Cora L | Amazon S | Amazon F | Amazon L | Coauthor S | Coauthor F | Coauthor L |
|---|---|---|---|---|---|---|---|---|---|---|
| MSP | No | 70.90 | 85.39 | 91.36 | 98.27 | 97.31 | **93.97** | 95.30 | 97.05 | 94.88 |
| ODIN | No | 49.92 | 49.88 | 49.80 | 93.24 | 81.15 | 65.97 | 52.14 | 51.54 | 51.44 |
| Mahalanobis | No | 46.68 | 49.93 | 67.62 | 71.69 | 76.50 | 73.25 | 80.46 | 93.23 | 85.36 |
| Energy | No | 71.73 | **86.15** | 91.40 | 98.51 | **97.87** | 93.81 | 96.18 | **97.88** | **95.87** |
| GKDE | No | 68.61 | 82.79 | 57.23 | 76.39 | 58.96 | 65.58 | 65.87 | 80.69 | 61.15 |
| GPN | No | **77.47** | 85.88 | 90.34 | 97.17 | 87.91 | 92.72 | 34.67 | 72.56 | 83.65 |
| GNNSAFE | No | 87.52 | 93.44 | 92.80 | 99.58 | 98.55 | 97.35 | 99.60 | 99.64 | 97.23 |
| OE | Yes | 67.98 | 81.83 | 89.47 | **99.60** | 98.39 | 95.39 | 97.86 | 99.04 | 96.04 |
| Energy FT | Yes | 75.88 | 88.15 | 91.36 | 98.83 | 98.55 | 97.35 | 98.84 | 99.43 | 96.23 |
| GNNSAFE++ | Yes | 90.62 | 95.56 | 92.75 | 99.82 | 99.64 | 97.51 | 99.99 | 99.97 | 97.89 |

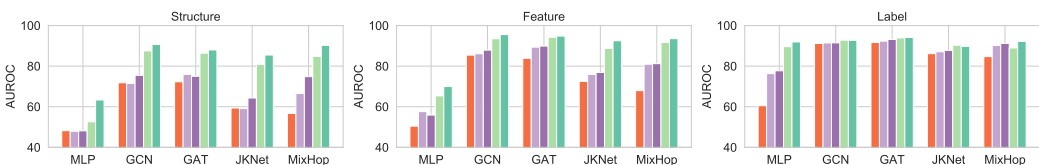

Figure 2: Performance comparison of MSP, Energy, Energy FT, GNNSAFE and GNNSAFE++ w.r.t. different encoder backbones on `Cora` with three OOD types.

Such results also verify our hypothesis in Section 3.2 that the structured propagation can leverage the topological information to boost OOD detection with data inter-dependence.

**How does the energy-regularized training help OOD detection?** We next investigate into the effectiveness of the introduced energy-regularized training. In comparison with GNNSAFE, we found that GNNSAFE++ indeed yields better relative performance across the experiments of Table 1 and 2, which is further verified by the visualization in Fig. 1 where we compare the energy scores of the models trained w/o and w/ the regularization loss (the middle and right columns, respectively) and see that GNNSAFE++ can indeed produce larger gap between in-distribution and OOD data. Furthermore, we notice that the performance gains achieved in Table 2 are not that considerable as Table 1. In particular, the AUROCs given by GNNSAFE and GNNSAFE++ are comparable on `Amazon` and `Coauthor`. The possible reason is that the OOD data on the multi-graph dataset `Twitch` and large temporal dataset `Arxiv` is introduced by the context information and the distribution shift may not be that significant, which makes it harder for discrimination, and the energy regularization can significantly facilitate OOD detection. On the other hand, the energy regularization on auxiliary OOD training data may not be necessary for cases (like other three datasets) where the distribution shifts between OOD and in-distribution data are intense enough.

### 4.3 ANALYSIS OF HYPER-PARAMETERS

We next analyze the impact of using different encoder backbones and hyper-parameters, to shed more lights on the practical efficacy and usage of our proposed model.

**Will different encoder backbones impact the detection results?** Fig. 2 compares the performance of MSP, Energy, Energy FT, GNNSAFE and GNNSAFE++ w.r.t. using different encoder backbones including MLP, GAT (Velickovic et al., 2018), JKNet (Xu et al., 2018), MixHop (Abu-El-Haija et al., 2019) on `Cora` with three OOD types. We found that different backbones indeed impact the detection performance and in particularly, GNN backbones bring up significantly better AUROC than MLP due to their stronger expressiveness for encoding topological features. However, we observe that the relative performance of five models are mostly consistent with the results in Table 2 where we use GCN as the backbone and two GNNSAFE models outperform other competitors in nearly all the cases. In the right figure, even with MLP backbone, two GNNSAFE models perform competitively as the counterparts with GNN backbones, which also validates the effectiveness of our energy propagation scheme that helps to leverage topological information over graph structures.

**How do the propagation depth and self-loop strength impact the inference-time performance?** Fig. 3(a) plots the results of GNNSAFE++ with propagation step $K$ increasing from

1 to 64 and self-loop strength $\alpha$ varing between 0.1 and 0.9. As shown in the figure, the AUROC scores first go up and then degrade as the propagation steps increase. The reason is that a moderate number of propagation steps can indeed help to leverage structural dependence for boosting OOD detection yet too large propagation depth could potentially overly smooth the estimated energy and make the detection scores indistinguishable among different nodes. Similar trend is observed for $\alpha$ which contributes to better performance with the value set as a moderate value, e.g., 0.5 or 0.7. This is presumably because too large $\alpha$ would counteract the effect of neighbored information and too small $\alpha$ would overly concentrate on the other nodes.

**How do the regularization weight and margins impact the training?** We discuss the performance variation of GNNSAFE++ w.r.t. the configurations of $t_{in}$, $t_{out}$ and $\lambda$ in Fig. 3(b) and (c). We found that the performance of GNNSAFE++ is largely dependent on $\lambda$ which controls the strength of energy regularization and its optimal values also vary among different datasets, though we merely tune it with a coarse searching space $\{0.01, 0.1, 1.0\}$. Besides, a proper setting for the margin $t_{in}$ and $t_{out}$ yields stably good performance while too large margin would lead to performance degradation.

## 5 RELATED WORKS

**Out-of-distribution detection for neural networks** Detecting OOD samples on which the models should have low confidence has been extensively studied by recent literature. From different technical aspects, prior art proposes to adopt confidence scores from pre-trained models (Hendrycks & Gimpel, 2016; Hendrycks et al., 2019a), add regularization on auxiliary OOD training data (Bevandić et al., 2018; Liu et al., 2020; Mohseni et al., 2020), or train a generative model for detecting OOD examples (Ren et al., 2019). However, these methods mostly assume instances (e.g., images) are i.i.d. generated, and ignore the widely existing problem scenario that involves inter-dependent data.

**Out-of-distribution detection on graphs** In contrast, OOD detection on graph data that contains inter-dependent nodes has remained as an under-explored research area. The recent work Li et al. (2022) proposes a flexible generative framework that involves node features, labels and graph structures and derives a posterior distribution for OOD detection. Moreover, Bazhenov et al. (2022) compares different OOD detection schemes from uncertainty estimation perspective. These works focus on graph classification where each instance itself is a graph and can be treated as i.i.d. samples as well. In terms of node classification where the inter-dependence of instances becomes non-negligible, Zhao et al. (2020) and Stadler et al. (2021) propose Bayesian GNN models that can detect OOD nodes on graphs. Different from these approaches, the OOD discriminator in our model is directly extracted from the predicted logits of standard GNNs trained with the supervised classification loss, which keeps a simple and general form that is agnostic to GNN architectures. Also, our model is applicable for off-the-shelf discriminative GNN classifiers and is free from training complicated graph generative models for density estimation or any extra OOD discriminator.

**Out-of-distribution generalization** Within the general picture of learning with distribution shifts, another related problem orthogonal to ours is to enhance the model's predictive performance on OOD testing data, i.e., OOD generalization (Muandet et al., 2013). The OOD data on graphs is of various types dependent on specific distribution shifts in different scenarios and datasets, including but not limited to cross-domain graph shifts and temporal graph shifts used for experiments in Wu et al. (2022a), subgroup attribute shifts in node classification (Ma et al., 2021; Bi et al., 2023), molecule size/scaffold shifts in molecular property prediction (Yang et al., 2022c), temporal context shifts in recommender systems (Yang et al., 2022a), feature space expansion in tabular predictive tasks (Wu et al., 2021), etc. As an exploratory work in OOD detection on graphs, we focus on the cross-domain shifts and temporal shifts in our experiments as two real-world scenarios, and one promising future direction is to extend the evaluation of OOD detection to other distribution shifts on graphs.

## 6 CONCLUSIONS

We have shown in this paper that graph neural networks trained with supervised loss can intrinsically be an effective OOD discriminator for detecting OOD testing data on which the model should avoid prediction. Our model is simple, general, and well justified by theoretical analysis. We also verify its practical efficacy through extensive experiments on real-world datasets with various OOD data types. We hope our work can empower GNNs for real applications where safety considerations matter.

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

## A  PROOFS FOR TECHNICAL RESULTS

### A.1  PROOF FOR PROPOSITION 1

*Proof.* The proof is kind of straight-forward as followed similar reasoning line of the section 3.1 in Liu et al. (2020). The gradient of $\mathcal{L}_{nll}$ w.r.t. $\theta$ is given by

$$
\begin{aligned}
\frac{\partial \mathcal{L}_{nll}}{\partial \theta} &= \sum_{i \in \mathcal{I}_{tr}} \left( -\frac{\partial h_\theta(\mathbf{x}_i, \mathcal{G}_{\mathbf{x}_i})_{[y_i]}}{\partial \theta} + \sum_{c=1}^{C} \frac{\partial h_\theta(\mathbf{x}_i, \mathcal{G}_{\mathbf{x}_i})_{[c]}}{\partial \theta} \frac{e^{h_\theta(\mathbf{x}_i, \mathcal{G}_{\mathbf{x}_i})_{[c]}}}{\sum_{c'=1}^{C} e^{h_\theta(\mathbf{x}_i, \mathcal{G}_{\mathbf{x}_i})_{[c']}}} \right) \\
&= \sum_{i \in \mathcal{I}_{tr}} \left( \frac{\partial E(\mathbf{x}_i, \mathcal{G}_{\mathbf{x}_i}, y_i)}{\partial \theta} - \sum_{c=1}^{C} \frac{\partial E(\mathbf{x}_i, \mathcal{G}_{\mathbf{x}_i}, c)}{\partial \theta} \frac{e^{h_\theta(\mathbf{x}_i, \mathcal{G}_{\mathbf{x}_i})_{[c]}}}{\sum_{c'=1}^{C} e^{h_\theta(\mathbf{x}_i, \mathcal{G}_{\mathbf{x}_i})_{[c']}}} \right) \\
&= (1 - p(y = y_i \mid \mathbf{x}_i, \mathcal{G}_{\mathbf{x}_i})) \frac{\partial E(\mathbf{x}_i, \mathcal{G}_{\mathbf{x}_i}, y_i; h_\theta)}{\partial \theta} - \sum_{c \neq y_i} p(y = c \mid \mathbf{x}_i, \mathcal{G}_{\mathbf{x}_i}) \frac{\partial E(\mathbf{x}_i, \mathcal{G}_{\mathbf{x}_i}, c; h_\theta)}{\partial \theta}.
\end{aligned}
\tag{10}
$$

We can see from the above equation that the training procedure overall minimizing the first-order gradient of $\mathcal{L}_{nll}$ will decrease the energy score $E(\mathbf{x}_i, \mathcal{G}_{\mathbf{x}_i}, y_i; h_\theta)$ and increase the energy $E(\mathbf{x}_i, \mathcal{G}_{\mathbf{x}_i}, c; h_\theta)$. In a general sense, the energy value produced by the trained model tends to become lower for any in-distribution instance $(\mathbf{x}, \mathcal{G}_{\mathbf{x}}, y) \sim \mathcal{D}_{in}$. □

## A.2 PROOF FOR PROPOSITION 2

*Proof.* If $\frac{\sum_j a_{ij} E_j^{(k-1)}}{\sum_j a_{ij}} < E_i^{(k-1)}$, we can re-write the update in Eqn. 7 as the scalar-form for each instance:

$$
\begin{aligned}
E_i^{(k)} &= \alpha E_i^{(k-1)} + (1 - \alpha) \frac{\sum_j a_{ij} E_j^{(k-1)}}{\sum_j a_{ij}} \\
&> \alpha E_i^{(k-1)} + (1 - \alpha) E_i^{(k-1)} \\
&= E_i^{(k-1)}.
\end{aligned}
\tag{11}
$$

The opposite result can be similarly proved for the case $\frac{\sum_j a_{ij} E_j^{(k-1)}}{\sum_j a_{ij}} > E_i^{(k-1)}$. $\quad\square$

## A.3 PROOF FOR PROPOSITION 3

*Proof.* We define $E(\mathbf{x}, \mathcal{G}_{\mathbf{x}}; h_{\theta*})$ as the energy yielded by the model $h_{\theta*}$ minimizing the regularization loss $\mathcal{L}_{reg}$. And, $h_{\theta\dagger}$ denotes the model yielding the optimal predictive softmax distribution that minimizes the supervised classification loss $\mathcal{L}_{sup}$, i.e.,

$$
\frac{e^{h_{\theta\dagger}(\mathbf{x}, \mathcal{G}_{\mathbf{x}})_{[y]}}}{\sum_{c=1}^{C} e^{h_{\theta\dagger}(\mathbf{x}, \mathcal{G}_{\mathbf{x}})_{[c]}}} = \operatorname{argmin}_{p(y|\mathbf{x}, \mathcal{G}_{\mathbf{x}})} \mathbb{E}_{(\mathbf{x}, \mathcal{G}_{\mathbf{x}}, y) \in \mathcal{D}_{in}}[-\log p(y|\mathbf{x}, \mathcal{G}_x)].
\tag{12}
$$

Notice that $E(\mathbf{x}, \mathcal{G}_{\mathbf{x}}; h_\theta) = -\log \sum_{c=1}^{C} e^{h_\theta(\mathbf{x}, \mathcal{G}_{\mathbf{x}})}$. Therefore we have

$$
\begin{aligned}
E(\mathbf{x}, \mathcal{G}_{\mathbf{x}}; h_{\theta*}) &= E(\mathbf{x}, \mathcal{G}_{\mathbf{x}}; h_{\theta*}) - E(\mathbf{x}, \mathcal{G}_{\mathbf{x}}; h_{\theta\dagger}) - \log \sum_{c=1}^{C} e^{h_{\theta\dagger}(\mathbf{x}, \mathcal{G}_{\mathbf{x}})} \\
&= -\log \left( e^{-E(\mathbf{x}, \mathcal{G}_{\mathbf{x}}; h_{\theta*}) + E(\mathbf{x}, \mathcal{G}_{\mathbf{x}}; h_{\theta\dagger})} \cdot \sum_{c=1}^{C} e^{h_{\theta\dagger}(\mathbf{x}, \mathcal{G}_{\mathbf{x}})} \right) \\
&= -\log \sum_{c=1}^{C} e^{h_{\theta\dagger}(\mathbf{x}, \mathcal{G}_{\mathbf{x}}) - E(\mathbf{x}, \mathcal{G}_{\mathbf{x}}; h_{\theta*}) + E(\mathbf{x}, \mathcal{G}_{\mathbf{x}}; h_{\theta\dagger})}.
\end{aligned}
\tag{13}
$$

The above relationship suggests that the energy $E(\mathbf{x}, \mathcal{G}_{\mathbf{x}}; h_{\theta*})$ is secretely equivalent to the energy induced by the predicted logits $h_{\theta\dagger}(\mathbf{x}, \mathcal{G}_{\mathbf{x}}) - E(\mathbf{x}, \mathcal{G}_{\mathbf{x}}; h_{\theta*}) + E(\mathbf{x}, \mathcal{G}_{\mathbf{x}}; h_{\theta\dagger})$. We can thereby denote the predictive softmax distribution of $E(\mathbf{x}, \mathcal{G}_{\mathbf{x}}; h_{\theta*})$ as

$$
\begin{aligned}
p(y|\mathbf{x}, \mathcal{G}_{\mathbf{x}}) &= \frac{e^{h_{\theta\dagger}(\mathbf{x}, \mathcal{G}_{\mathbf{x}})_{[y]} - E(\mathbf{x}, \mathcal{G}_{\mathbf{x}}; h_{\theta*}) + E(\mathbf{x}, \mathcal{G}_{\mathbf{x}}; h_{\theta\dagger})}}{\sum_{c=1}^{C} e^{h_{\theta\dagger}(\mathbf{x}, \mathcal{G}_{\mathbf{x}})_{[c]} - E(\mathbf{x}, \mathcal{G}_{\mathbf{x}}; h_{\theta*}) + E(\mathbf{x}, \mathcal{G}_{\mathbf{x}}; h_{\theta\dagger})}} \\
&= \frac{e^{h_{\theta\dagger}(\mathbf{x}, \mathcal{G}_{\mathbf{x}})_{[y]}}}{\sum_{c=1}^{C} e^{h_{\theta\dagger}(\mathbf{x}, \mathcal{G}_{\mathbf{x}})_{[c]}}}.
\end{aligned}
\tag{14}
$$

The above predictive distribution exactly minimizes $\mathcal{L}_{sup}$ as defined by Eqn. 12. We thus have proven that the optimal energy that minimizes $\mathcal{L}_{reg}$ intrinsically induce the predictive softmax distribution that minimizes $\mathcal{L}_{sup}$. On the other hand, given our assumption that the optimal energy function can well distinguish the scores for in-distribution and OOD inputs, we thus obtain proper choices for the regularization loss $\mathcal{L}_{reg}$ whose optimal energy satisfies the two requirements mentioned upfront in Section 3.3.

$\quad\square$

## B EXPERIMENTAL DETAILS

We supplement experiment details for reproducibility. Our implementation is based on Ubuntu 16.04.6, Cuda 10.2, Pytorch 1.9.0 and Pytorch Geometric 2.0.3. Most of the experiments are running with a NVIDIA 2080Ti with 11GB memory, except that for cases where the model requires larger GPU memory we use a NVIDIA 3090 with 24GB memory for experiments.

## B.1 DATASET INFORMATION

The datasets used in our experiment are all public available as common benchmarks for evaluating graph learning models. For `ogbn-Arxiv`, we use the preprocessed dataset and data loader provided by the OGB package[2]. For other datasets, we use the data loader provided by the Pytorch Geometric package[3].

`Cora` (Sen et al., 2008) is a citation network where each node denotes a published paper and each edge indicates the citation relationship. The goal is to predict each paper's topic as the label class. This dataset contains 2,708 nodes, 5,429 edges, 1,433 features and 7 classes. We use three different ways for synthetically creating OOD data as mentioned in Section 4.1. For training/validation/testing splits on in-distribution data, we use follow the semi-supervised learning setting by Kipf & Welling (2017) and use its provided splits.

`Amazon-Photo` (McAuley et al., 2015) is an item co-purchasing network on Amazon where each node denotes a product and each edge indicates that two linked products are frequently purchased together. The node label denotes the category of the product. This dataset contains 7,650 nodes, 238,162 edges, 745 features, and 8 classes. Similar to `Cora`, we use three ways for synthetically creating OOD data. And, for in-distribution data, we follow the common practice (Kipf & Welling, 2017) and use random splits with 1:1:8 for training/validation/testing.

`Coauthor-CS` (Sinha et al., 2015) is a coauthor network of computer science. Nodes denote authors and they are connected as edges if the two authors co-authored a paper. The goal is to predict the respective field of study for authors based on their papers' keywords as node features. This dataset contains 18,333 nodes, 163,788 edges, 6,805 features, and 15 classes. We also use the aforementioned three ways to construct OOD data and for in-distribution data, we follow the common practice using random splits with 1:1:8 for training/validation/testing.

`Twitch-Explicit` (Rozemberczki & Sarkar, 2021) is a multi-graph dataset, where each subgraph corresponds to a social network in one region. Nodes in this dataset represent game players on Twitch and edges indicate two users follow each other. Node features are embeddings of games played by the Twitch users, and the label class indicates whether a user streams mature content. The node numbers of the subgraphs are ranged from 1,912 to 9,498, with edge numbers from 31,299 to 153,138 and shared feature dimension 2,545. As mentioned in Section 4.1 we use subgraph DE as in-distribution data where we use random splits with 1:1:8 for training/validation/testing. Besides, subgraph ENGB is used as OOD exposure and ES, FR, RU are used for OOD testing data.

`ogbn-Arxiv` (Hu et al., 2020) is a relatively large graph dataset that records the citation information from 1960 to 2020. Each node denotes a paper with a subject area as the label to predict. Edges indicate citation relationship and each node has a 128-dimensional feature vector obtained by the word embeddings of its title and abstract. As described in Section 4.1 we use the time information for partitioning in-distribution and OOD data. For the in-distribution portion, we also use 1:1:8 random splits for training/validation/testing. Notice that here our partition for the in-distribution and OOD data disables us to use the original splitting by Hu et al. (2020) which used time information for splitting training and testing nodes. In our setting, the random splitting also guarantees that the training, validation and testing nodes (of in-distribution data) come from an identical distribution that is distinct from that of OOD data.

## B.2 IMPLEMENTATION DETAILS

**Architectures** As mentioned in Section 4.1, our basic encoder backbone is set as a GCN, and we also consider other models such as MLP, GAT, JKNet and MixHop for further discussions. We summary the architectural information of all the encoder backbones used in our experiments as follows.

- GCN: 2 GCNConv layers with hidden size 64, ReLU activation, self-loop and batch normalization.
- MLP: 2 fully-connected layers with hidden size 64 and ReLU activation.

---

[2]https://github.com/snap-stanford/ogb
[3]https://pytorch-geometric.readthedocs.io/en/latest/modules/datasets.html

- GAT: 2 GATConv layers with hidden size 64, ELU activation, head number $[2, 1]$ and batch normalization.
- JKNet: 2 GCNConv layers with hidden size 64, ReLU activation, self-loop and batch normalization. The jumping knowledge module is using max pooling.
- MixHop: 2 MixHop layers with hop number 2, hidden size 64, ReLU activation, and batch normalization.

**Hyper-parameters**  As described in Section 4.1, we consider the following hypermeters as a default setting: the propagation step $K = 2$ and the propagation self-loop weight $\alpha = 0.5$. For GNNSAFE, we only consider grid-search for the learning rate within $\{0.1, 0.01, 0.001\}$ using validation set. For GNNSAFE++, we use the validation set for additionally search for margin hyper-parameters $t_{in} \in \{-9, -7, -5\}$, $t_{out} \in \{-1, -2, -3, -4\}$ and the regularization weight $\lambda \in \{0.01, 0.1, 1.0\}$.

**Training Details**  The model GNNSAFE is trained with the supervised classification loss on labeled training data, and GNNSAFE++ is trained with the supervised loss plus the regularization loss on the OOD training data. We use the supervised classification loss (i.e., cross-entropy) on validation set as the indicator of epoch selection for reporting the testing result. More specifically, in each run, we train the model with 200 epochs as a fixed budget and report the testing performance produced by the epoch where the model yields the lowest classification loss on validation data.

**Competitors**  For competitor models, we use their implementations provided by the original papers and do some adaptation for our setting. As mentioned in Section 4.1, we use the same encoder backbone GCN (with the same hidden size 64 and layer number 2) for baselines MSP, ODIN, Mahalanobis, Energy and Energy FT. For GKDE and GPN, we use their public implementation, refer to their hyper-parameter settings reported in the paper and do some fine-tuning for different datasets.

### B.3 EVALUATION METRICS

We provide more details concerning the evaluation metrics we used in our experiment. We follow the common practice of OOD detection literature and use AUROC, AUPR and FPR95 for evaluating the performance for detecting OOD data in the testing set. These metrics are independent of the threshold $\tau$. Specifically, AUROC refers to the Area Under the Receiver Operating Characteristic (ROC) curve. The ROC curve presents a trade-off between the true positive rate and the false positive rate under different thresholds in $0 \sim 1$. Intuitively, the AUROC can be understood as the probability that for any pair of a positive example (i.e., in-distribution sample) and a negative example (out-of-distribution sample), the positive one has a larger estimated score than the negative. However, AUROC is not always an ideal metric, especially when the positive class and negative class are imbalanced. Under such a case, the AUPR adjusts for different positive/negative base rates. AUPR refers to the Area Under the PR curve which presents the trade-off between the precision and recall under different thresholds in $0 \sim 1$. Furthermore, FPR95 is also a widely used metric which refers to the False Positive Rate at 95% true positive rate (TPR). FPR95 can be interpreted as the probability that a out-of-distribution sample is misclassified as in-distribution when the TPR is as high as 95%.

## C  ADDITIONAL EXPERIMENT RESULTS

We supplement more experimental results in this section. In specific, we report detailed OOD detection performance on each OOD dataset (different subgraphs for `Twitch` and different years for `Arxiv`) in Table 3 and 4 as complementary for Table 1 in the maintext. Besides, in Table 5, 6 and 7 we report the AUROC/AUPR/FPR95 and in-distribution testing accuracy on `Cora`, `Amazon` and `Coauthor` as complementary for Table 2 in the maintext. In Table 8 we compare the training time per epoch and inference time of our model and the competitors. Moreover, we compare the training time per epoch and inference time of GNNSAFE++ and the competitors in Table 8. Also, we present more hyper-parameter analysis results in Fig. 3 including $\alpha$, $K$, $t_{in}$, $t_{out}$ and $\lambda$.

Table 3: Out-of-distribution detection performance measured by AUROC(↑)/AUPR(↑)/FPR95(↓) on OOD sub-graphs ES, FR and RU of `Twitch` dataset.

| Model | OOD Expo | Twitch-ES | | | Twitch-FR | | | Twitch-RU | | |
|---|---|---|---|---|---|---|---|---|---|---|
| | | AUROC | AUPR | FPR95 | AUROC | AUPR | FPR95 | AUROC | AUPR | FPR95 |
| MSP | No | 37.72 | 53.08 | 98.09 | 21.82 | 38.27 | 99.25 | 41.23 | 56.06 | 95.01 |
| ODIN | No | 83.83 | 80.43 | 33.28 | 59.82 | 64.63 | 92.57 | 58.67 | 72.58 | 93.98 |
| Mahalanobis | No | 45.66 | 58.82 | 95.48 | 40.40 | 46.69 | 95.54 | 55.68 | 66.42 | 90.13 |
| Energy | No | 38.80 | 54.26 | 95.70 | 57.21 | 61.48 | 91.57 | 57.72 | 66.68 | 87.57 |
| GKDE | No | 48.70 | 61.05 | 95.37 | 49.19 | 52.94 | 95.04 | 46.48 | 62.11 | 95.62 |
| GPN | No | 53.00 | 64.24 | 95.05 | 51.25 | 55.37 | 93.92 | 50.89 | 65.14 | 99.93 |
| OE | Yes | 55.97 | 69.49 | 94.94 | 45.66 | 54.03 | 95.48 | 55.72 | 70.18 | 95.07 |
| Energy FT | Yes | 80.73 | 87.56 | 76.76 | 79.66 | 81.20 | 76.39 | 93.12 | 95.36 | 30.72 |
| GNNSAFE | No | 49.07 | 57.62 | 93.98 | 63.49 | 66.25 | 90.80 | 87.90 | 89.05 | 43.95 |
| GNNSAFE++ | Yes | 94.54 | 97.17 | 44.06 | 93.45 | 95.44 | 51.06 | 98.10 | 98.74 | 5.59 |

Table 4: Out-of-distribution detection performance measured by AUROC(↑)/AUPR(↑)/FPR95(↓) on OOD datasets of papers published in 2018, 2019 and 2020, respectively, on `Arxiv`.

| Model | OOD Expo | Arxiv-2018 | | | Arxiv-2019 | | | Arxiv-2020 | | |
|---|---|---|---|---|---|---|---|---|---|---|
| | | AUROC | AUPR | FPR95 | AUROC | AUPR | FPR95 | AUROC | AUPR | FPR95 |
| MSP | No | 61.66 | 70.63 | 91.67 | 63.07 | 66.00 | 90.82 | 67.00 | 90.92 | 89.28 |
| ODIN | No | 53.49 | 63.06 | 100.0 | 53.95 | 56.07 | 100.0 | 55.78 | 87.41 | 100.0 |
| Mahalanobis | No | 57.08 | 65.09 | 93.69 | 56.76 | 57.85 | 94.01 | 56.92 | 85.95 | 95.01 |
| Energy | No | 61.75 | 70.41 | 91.74 | 63.16 | 65.78 | 90.96 | 67.70 | 91.15 | 89.69 |
| GKDE | No | 56.29 | 66.78 | 94.31 | 57.87 | 62.34 | 93.97 | 60.79 | 88.74 | 93.31 |
| GPN | No | - | - | - | - | - | - | - | - | - |
| OE | Yes | 67.72 | 75.74 | 86.67 | 69.33 | 72.15 | 85.52 | 72.35 | 92.57 | 83.28 |
| Energy FT | Yes | 69.58 | 76.31 | 82.10 | 70.58 | 72.03 | 81.30 | 74.53 | 93.08 | 78.36 |
| GNNSAFE | No | 66.47 | 74.99 | 89.44 | 68.36 | 71.57 | 88.02 | 78.35 | 94.76 | 83.57 |
| GNNSAFE++ | Yes | 70.4 | 78.62 | 81.47 | 72.16 | 75.43 | 79.33 | 81.75 | 95.57 | 71.50 |

Table 5: Out-of-distribution detection performance measured by AUROC(↑)/AUPR(↑)/FPR95(↓) on `Cora` with three OOD types (structure manipulation, feature interpolation, label leave-out).

| Model | Cora-Structure | | | | Cora-Feature | | | | Cora-Label | | | |
|---|---|---|---|---|---|---|---|---|---|---|---|---|
| | AUROC | AUPR | FPR95 | ID ACC | AUROC | AUPR | FPR95 | ID ACC | AUROC | AUPR | FPR95 | ID ACC |
| MSP | 70.90 | 45.73 | 87.30 | 75.50 | 85.39 | 73.70 | 64.88 | 75.30 | 91.36 | 78.03 | 34.99 | 88.92 |
| ODIN | 49.92 | 27.01 | 100.0 | 74.90 | 49.88 | 26.96 | 100.0 | 75.00 | 49.80 | 24.27 | 100.0 | 88.92 |
| Mahalanobis | 46.68 | 29.03 | 98.19 | 74.90 | 49.93 | 31.95 | 99.93 | 74.90 | 67.62 | 42.31 | 90.77 | 88.92 |
| Energy | 71.73 | 46.08 | 88.74 | 76.00 | 86.15 | 74.42 | 65.81 | 76.10 | 91.40 | 78.14 | 41.08 | 88.92 |
| GKDE | 68.61 | 44.26 | 84.34 | 73.70 | 82.79 | 66.52 | 68.24 | 74.80 | 57.23 | 27.50 | 88.95 | 89.87 |
| GPN | 77.47 | 53.26 | 76.22 | 76.50 | 85.88 | 73.79 | 56.17 | 77.00 | 90.34 | 77.40 | 37.42 | 91.46 |
| OE | 67.98 | 46.93 | 95.31 | 71.80 | 81.83 | 70.84 | 83.79 | 73.30 | 89.47 | 77.01 | 46.55 | 87.97 |
| Energy FT | 75.88 | 49.18 | 67.73 | 75.50 | 88.15 | 75.99 | 47.53 | 75.30 | 91.36 | 78.49 | 37.83 | 90.51 |
| GNNSAFE | 87.52 | 77.46 | 73.15 | 75.80 | 93.44 | 88.19 | 38.92 | 76.40 | 92.80 | 82.21 | 30.83 | 88.92 |
| GNNSAFE++ | 90.62 | 81.88 | 53.51 | 76.10 | 95.56 | 90.27 | 27.73 | 76.80 | 92.75 | 82.64 | 34.08 | 91.46 |

Table 6: Out-of-distribution detection performance measured by AUROC(↑)/AUPR(↑)/FPR95(↓) on `Amazon` with three OOD types (structure manipulation, feature interpolation, label leave-out).

| Model | Amazon-Structure | | | | Amazon-Feature | | | | Amazon-Label | | | |
|---|---|---|---|---|---|---|---|---|---|---|---|---|
| | AUROC | AUPR | FPR95 | ID ACC | AUROC | AUPR | FPR95 | ID ACC | AUROC | AUPR | FPR95 | ID ACC |
| MSP | 98.27 | 98.54 | 6.13 | 92.84 | 97.31 | 95.16 | 8.72 | 92.89 | 93.97 | 91.32 | 26.65 | 95.76 |
| ODIN | 93.24 | 95.26 | 65.44 | 92.84 | 81.15 | 78.47 | 100.0 | 92.71 | 65.97 | 57.80 | 90.23 | 96.08 |
| Mahalanobis | 71.69 | 79.01 | 99.91 | 92.79 | 76.50 | 71.14 | 76.12 | 92.86 | 73.25 | 66.89 | 74.30 | 95.76 |
| Energy | 98.51 | 98.72 | 4.97 | 92.86 | 97.87 | 95.64 | 6.00 | 92.96 | 93.81 | 91.13 | 28.48 | 95.72 |
| GKDE | 76.39 | 81.58 | 99.25 | 87.57 | 58.96 | 66.76 | 99.28 | 86.18 | 65.58 | 65.20 | 96.87 | 89.37 |
| GPN | 97.17 | 96.39 | 11.65 | 88.51 | 87.91 | 84.77 | 49.11 | 90.05 | 92.72 | 90.34 | 37.16 | 90.07 |
| OE | 99.60 | 99.61 | 0.51 | 92.61 | 98.39 | 96.24 | 4.34 | 92.30 | 95.39 | 92.53 | 17.72 | 95.72 |
| Energy FT | 98.83 | 99.14 | 1.31 | 92.79 | 98.68 | 96.82 | 2.84 | 92.52 | 96.61 | 94.92 | 13.78 | 94.83 |
| GNNSAFE | 99.58 | 99.76 | 0.00 | 92.53 | 98.55 | 98.99 | 0.31 | 92.81 | 97.35 | 97.12 | 6.59 | 95.76 |
| GNNSAFE++ | 99.82 | 99.89 | 0.00 | 92.22 | 99.64 | 99.68 | 0.13 | 92.39 | 97.51 | 97.07 | 6.18 | 95.84 |

Table 7: Out-of-distribution detection performance measured by AUROC(↑)/AUPR(↑)/FPR95(↓) on `Coauthor` with three OOD types (structure manipulation, feature interpolation, label leave-out).

| Model | Coauthor-Structure | | | | Coauthor-Feature | | | | Coauthor-Label | | | |
|---|---|---|---|---|---|---|---|---|---|---|---|---|
| | AUROC | AUPR | FPR95 | ID ACC | AUROC | AUPR | FPR95 | ID ACC | AUROC | AUPR | FPR95 | ID ACC |
| MSP | 95.30 | 94.37 | 24.75 | 92.47 | 97.05 | 96.93 | 15.55 | 92.45 | 94.88 | 97.99 | 23.81 | 95.18 |
| ODIN | 52.14 | 48.83 | 99.92 | 92.34 | 51.54 | 45.50 | 100.0 | 92.39 | 51.44 | 74.79 | 100.0 | 95.15 |
| Mahalanobis | 80.46 | 76.65 | 70.75 | 92.33 | 93.23 | 90.88 | 28.10 | 92.34 | 85.36 | 93.61 | 45.41 | 95.19 |
| Energy | 96.18 | 95.25 | 18.02 | 92.75 | 97.88 | 97.69 | 9.75 | 92.75 | 95.87 | 98.34 | 18.69 | 95.20 |
| GKDE | 65.87 | 72.65 | 99.48 | 88.62 | 80.69 | 86.47 | 96.57 | 84.72 | 61.15 | 81.39 | 94.60 | 89.05 |
| GPN | 34.67 | 40.21 | 99.57 | 89.45 | 81.77 | 80.56 | 74.46 | 87.05 | 93.24 | 97.55 | 34.78 | 91.68 |
| OE | 97.86 | 96.81 | 9.23 | 92.60 | 99.04 | 98.80 | 4.44 | 92.64 | 96.04 | 98.50 | 18.17 | 95.10 |
| Energy FT | 98.84 | 97.78 | 3.97 | 92.61 | 99.43 | 99.25 | 2.25 | 92.50 | 96.23 | 98.51 | 17.07 | 95.20 |
| GNNSAFE | 99.60 | 99.69 | 0.26 | 92.73 | 99.64 | 99.66 | 0.51 | 92.73 | 97.23 | 98.98 | 12.06 | 95.21 |
| GNNSAFE++ | 99.99 | 99.99 | 0.02 | 92.92 | 99.97 | 99.95 | 0.09 | 92.87 | 97.89 | 99.24 | 9.43 | 95.24 |

Table 8: Comparison of training time per epoch and inference time of all the models on five datasets.

| Model | Cora | | Amazon | | Coauthor | | Twitch | | Arxiv | |
|---|---|---|---|---|---|---|---|---|---|---|
| | TR (s) | IN (s) | TR (s) | IN (s) | TR (s) | IN (s) | TR (s) | IN (s) | TR (s) | IN (s) |
| MSP | 0.008 | 0.013 | 0.011 | 0.023 | 0.060 | 0.166 | 0.008 | 0.040 | 0.043 | 0.173 |
| ODIN | 0.007 | 0.025 | 0.009 | 0.053 | 0.055 | 0.311 | 0.008 | 0.081 | 0.018 | 0.010 |
| Mahalanobis | 0.008 | 0.251 | 0.013 | 0.306 | 0.073 | 1.081 | 0.009 | 0.006 | 0.024 | 0.034 |
| Energy | 0.013 | 0.016 | 0.016 | 0.022 | 0.114 | 0.160 | 0.011 | 0.048 | 0.063 | 0.192 |
| GKDE | 0.007 | 0.015 | 0.008 | 0.025 | 0.069 | 0.236 | 0.006 | 0.043 | 0.030 | 0.171 |
| GPN | 10.02 | 40.30 | 6.480 | 25.86 | 28.58 | 114.44 | 2.850 | 13.88 | - | - |
| OE | 0.014 | 0.015 | 0.018 | 0.024 | 0.118 | 0.165 | 0.012 | 0.044 | 0.077 | 0.252 |
| Energy FT | 0.013 | 0.018 | 0.018 | 0.024 | 0.131 | 0.171 | 0.016 | 0.052 | 0.069 | 0.218 |
| GNNSAFE | 0.015 | 0.015 | 0.015 | 0.023 | 0.115 | 0.164 | 0.011 | 0.049 | 0.062 | 0.219 |
| GNNSAFE++ | 0.015 | 0.019 | 0.015 | 0.027 | 0.141 | 0.189 | 0.016 | 0.052 | 0.075 | 0.239 |

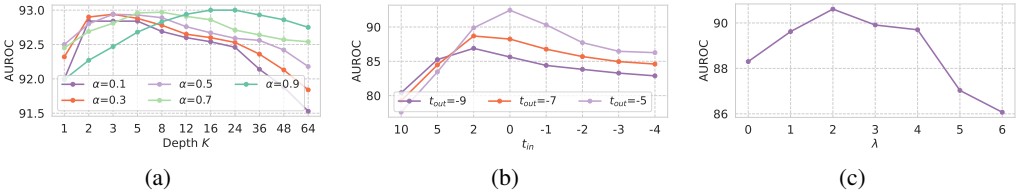

(a)        (b)        (c)

Figure 3: (a) Hyper-parameter analysis for propagation steps $K$ and self-loop strength $\alpha$. (b) Impact of margin hyper-parameter $t_{in}$ and $t_{out}$. (c) Impact of weight hyper-parameter $\lambda$ for energy regularization of GNNSAFE++. We `Cora` with structure manipulation for experiments.

