# OpenReview forum: "Energy-based Out-of-Distribution Detection for Graph Neural Networks"
_ICLR.cc/2023/Conference — ICLR 2023 poster_

### Official Review · Reviewer_PiYE · 2022-10-16

**Confidence:** 5
**Correctness:** 3
**Technical Novelty And Significance:** 2
**Empirical Novelty And Significance:** 3
**Recommendation:** 8

**Clarity, Quality, Novelty And Reproducibility:**

The writeup of the paper is very clear and the paper is easy to follow. The hyperparameter details along with the empirical evidence presented suggest that the results are likely to be reproducible. Though there are concerns regarding the novelty of the work as the idea of energy based models has been extensively utilized in the same literature of out of distribution detection and with similar formulations.


**Strength And Weaknesses:**

1. The experiments clearly demonstrate that the proposed approach is highly effective. It is very interesting that the GNN-safe model alone can provide substantially high detection performance.
2. The experiments in table 8 comparing the runtime of all the methods are further useful in demonstrating the effectiveness of the proposed approach.
4. Eq 11 in the appendix should have a sum instead of difference.
5. The authors mention in the last paragraph of page 4 that - “In specific, the generation of a node is conditioned on its neighbors, and thereby an in-distribution node tends to connect with other nodes that are sampled from in-distribution, and vice versa”. It would be more useful to provide some citations for this. Though they mention in the subsequent paragraph that the labels and features for the neighbors can be different, in such heterophilous graphs, the opposite trend may also hold.
6. There is strong emphasis that the energy of in-distribution samples tend to be lower than the out of distribution samples, however, the opposite trend emerges in figure 1 (a). Do the authors have some intuition of why this is happening?
7. There is no mention of the selection of the hyperparameter $\tau$, eq 8. Details and study of this hyperparameter will be highly relevant.


**Summary Of The Paper:**

The authors propose a new mechanism based on the energy-models for the task of out of distribution detection on graphs. They show that the graph neural networks trained with supervised loss objective can be intrinsically effective to detect OOD data on which the model should avoid prediction. The energy based method proposed in this work can be directly extracted through simple transformation from the predicted logits of a GNN classifier. Furthermore, the proposed method is agnostic to the choice of the underlying architecture used making it highly generalizable. This is further enhanced using a simple propagation scheme, following the popular label propagation framework, that can provably improve the detection performance by increasing the margin between the in-distribution and out of distribution. Lastly, they propose a direct extension of this by utilizing the energy based scheme in an auxiliary loss regularizer that can be trained along the supervised NLL objective. Experiments performed over a diverse collection of datasets and against SOTA methods clearly demonstrate the proposed method is highly effective.


**Summary Of The Review:**

Based on the questions, comments and concerns raised in the weakness section as well as the issue concerning novelty of the work, I lean towards weak rejection of the work.

---

> ### Author Response · Authors · 2022-11-10
> **Response to Reviewer PiYE**
>
> Thank you for the thorough review and constructive suggestions. We are encouraged to see that you appreciated the strong empirical performance and effectiveness of our model.
>
> > ***Q1: "Eq.11 in the appendix should have a sum instead of difference."***
>
> Thanks for pointing this out, we have fixed it in the revision.
>
> > ***Q2: "It would be more useful to provide some citations for this... in such heterophilous graphs, the opposite trend may also hold."***
>
> Thank you for the nice suggestion and thoughtful question which can help us to improve the presentation. In fact, the mentioned statement in our draft is our hypothesis/conjecture and we have modified the words to make it clearer in the revision. Such a hypothesis stems from an observation that the input graph reflects the underlying geometry behind data samples whose generation can be treated as dependent of the connected samples (Belkin et al., 2006), so it is natural to assume that the inter-connected nodes are generated from similar distributions. This motivates us to design the energy-based belief propagation to boost OOD detection.
>
> This hyphothesis is in fact compatible with the heterophily case where connected nodes can have different labels or features. Notice that here we focus on data distribution that is a joint distribution of node labels and features, so even if their data distributions are similar, two connected nodes can have different labels/features. This argument as well as our hypothesis are also validated by our empirical results which show that our model GNN-Safe with the energy propagation consistently outperforms Energy (the counterpart not using the propagation) on both Arxiv (homophily graph) and Twitch (heterophily graph).
>
> > ***Q3: "however, the opposite trend emerges in figure 1 (a)."***
>
> After double checking, we apologize that the original visualization in Fig.1(a) is wrong and the true energy should be the opposite. We have fixed it in the revision. With the correct figure, we can see that the trend is indeed consistent with our theory and claims in the paper.
>
> > ***Q4: "There is no mention of the selection of the hyperparameter $\tau$"***
>
> The hyper-parameter $\tau$ is not needed in evaluation, and only needs to be specified in practical situations where one expects the model to return the indication of OOD, i.e., 0 or 1 result. In our experiments, we measure the OOD detection by the threshold-independent metrics AUROC, AUPR and FPR. These metrics are agnostic of specific $\tau$'s and also widely used as common metrics by the literature. We have added more explaination about these metrics in Appendix B.3.
>
> > ***Q5: " the idea of energy based models has been extensively utilized in the same literature of out of distribution detection"***
>
> While energy-based models for OOD detection have been explored, we believe that the technical/empirical contributions are significant with the justifications below.
>
> - First, we study a new problem, OOD detection for data with inter-dependence, which is under-explored by the OOD detection community and fundamentally distinct from the setting of existing models (including the energy-based) that assume data samples are i.i.d. generated. In this sense, it requires non-trivially distinct considerations for modeling the non-i.i.d. data-generative nature for OOD detection, which is largely overlooked by existing models.
> - Second, even compared with existing energy models for OOD detection, our model is not a trivial adaptation from vision data (CNN) to graph data (GNN). From architectural view, existing methods in vision use CNN to individually process and model each x (an image), while our model leverages GNN to model the inter-dependence of instances x's (nodes in a graph). Furthermore, we propose a novel non-learning-based energy propogation scheme to boost the OOD detection, which indeed brings up significant gain in all the cases of our experiments. Besides, our theoretical insights in Sec 3.2 and 3.3 are also valuable to the community.
> - On empirical side, it also takes non-trivial efforts to make the energy model work smoothly in graph data that have distinct data format (often with categorical features) than images (continuous pixels). Given that this technical path has never been done for graph-structured data, we thus believe that our model implementation and empirical improvements are significant to the community.
>
> [1] Belkin et al., Manifold regularization: A geometric framework for learning from labeled and unlabeled examples. Journal of machine learning research, 2006

---

> > ### Comment · Reviewer_PiYE · 2022-11-13
> > **Response to the Authors**
> >
> > I appreciate the authors' effort in the improving the manuscript and clarifying the aforementioned questions/concerns. I am thus increasing my score.

---

### Official Review · Reviewer_2RDq · 2022-10-24

**Confidence:** 4
**Correctness:** 2
**Technical Novelty And Significance:** 2
**Empirical Novelty And Significance:** 2
**Recommendation:** 5

**Clarity, Quality, Novelty And Reproducibility:**

The clarity and quality are good. However, considering the previous work of Liu et al. 2020, the novelty is not significant enough.

**Strength And Weaknesses:**

Strength:
1. The paper is well-written and the motivation is clear.
2. The numerical results showed that the proposed two methods especially GNN-Safe-r are much better than the baselines.

Weakness:
1. The main idea is almost the same as that of Energy FT (Liu et al. 2020). It seems that the major difference is the proposed methods are for graph neural network rather than MLP and CNN.

2. In the experiments, what is the main difference in terms of the implementation between Energy FT and the proposed GNN-Safe-r?

3. One can observe that in many cases, GNN-Safe was outperformed by Energy FT, which verified the importance of OOD exposure in the training stage. Thus, one may train a binary classifier to classify the in-distribution samples and out-distribution samples into two different classes, which should have a good performance in the testing stage.

4. The OOD exposure violates the principle of OOD detection because it becomes a supervised learning problem with OOD labels.

**Summary Of The Paper:**

This paper proposed an energy-based out-of-distribution detection for graph neural network. The contributions include introducing energy-based method to graph out-of-distribution detection, providing some theoretical analysis, presenting energy propagation, and conducting a lot of experiments.

**Summary Of The Review:**

It seems that the main improvement over existing work such as Liu et al 2020 is that the proposed method focuses on GNN. Thus the contribution is not that significant. On the other hand, the numerical results indicate that OOD exposure plays a very important role in the success of the proposed method. It is not clear whether a direct binary classification between in-distribution and out-distribution can yield good OOD performance or not.

---

> ### Author Response · Authors · 2022-11-10
> **Response to Reviewer 2RDq (Part 1 of 2)**
>
> Thank you for the time and valuable feedback. We notice that your concerns lie in two major points: the novelty compared to (Liu et al. 2020) and the role of OOD exposure. Based on the review comments, we conjecture there might be some potential misunderstandings regarding our problem setting and technical contributions. Below is the detailed response.
>
> ### **Novelty**
>
> > ***Q1: "It seems that the major difference is the proposed methods are for graph neural network rather than MLP and CNN."***
>
> We respectfully disagree with this interpretation which might stem from potential misunderstandings on our problem setting. Existing methods for OOD detection, including (Liu et al. 2020), assume input data are independently sampled, so that each x (an image) is individually modeled by the CNN to yield an output prediction. In our setting, input data (nodes in a graph) have inter-connection and cannot be treated independently. For each x (a node), it is dependent on other x's (neighbored nodes) for data-generative modelling, and the GNN is not to process an invidual x but a whole dataset of x's. Therefore, our model is **not** a direct adaptation where the x changes from images to graphs and the network changes from CNN to GNN.
>
> Compared with existing energy models, e.g., (Liu et al. 2020), our proposed model also has clear difference. GNN-Safe introduces a non-learning-based scheme, i.e., energy-based belief propagation, which is specially designed for OOD detection on data with inter-dependence and indeed brings up significant performance gains. The results are consistently highlighted by Table 1, 2 where GNN-Safe (resp. GNN-Safe-r) outperforms the counterpart Energy (resp. Energy FT) without such a scheme by 30.4%/22.0% (resp. 12.8%/19.4%) on Twitch/Cora-Structure. Furthermore, our theorectical insights in Sec 3.2 and 3.3 are also new to the community, which justify the proposed energy propagation and energy-regularized learning (for GNN-Safe-r).
>
> > ***Q2: "What is the main difference in terms of the implementation between Energy FT and the proposed GNN-Safe-r?"***
>
> From model architectural view, again our model is **not** a trivial adaptation from CNN to GNN. In Energy FT, the CNN backbone indivially processes each x (an image), while in GNN-Safe(-r), the GNN is to model a batch of x's (nodes in a graph) that have inter-dependence. Also, as mentioned above, GNN-Safe(-r) additionally introduces energy belief propagation. Beyond these model-level differences, since the nature and characteristics of graph data are clearly different from images, e.g., distinct input feature space (categorical features for nodes in graph v.s. continuous pixels for images), it also takes non-trivial efforts to make the energy model work smoothly in practice under such an unexplored problem.
>
>
> ### **The role of OOD exposure**
>
>
> > ***Q3: "One can observe that in many cases, GNN-Safe was outperformed by Energy FT, which verified the importance of OOD exposure in the training stage"***
>
> We agree with the description for our experimental results yet respectfully have a disagreement on the inproper interpretation. It is **not surprising** to see that in some cases Energy FT achieves better scores than GNN-Safe since it uses **extra** OOD training data. To be specific, there are two situations considered by us w.r.t. OOD detection (which are also commonly studied by peer works in vision, e.g., [1-6]):
>
> - **S1**: Train a model with pure in-distribution data and test it for OOD detection on OOD data.
> - **S2**: Train a model with a mix of in-distribution and OOD data (the OOD exposure) and test it for OOD detection on new unseen OOD data (whose distribution is also different from the training OOD exposure).
>
> These two settings correspond to different models (as indicated by the second columns in our Table 1/2 ) and need separate comparisons (we thank Reviewer QtVR for highlighting their differences). Given these facts, the effectiveness of our approach is well supported by the empricial results. First, under the setting of S1, GNN-Safe significantly outperform peer competitors throughout all cases in Table 1 and 2. Second, under the setting of S2, GNN-Safe-r significantly outperform other baselines in all cases. Therefore, it is reasonable that Energy FT achieves better scores than GNN-Safe in some cases in consideration of that it uses extra OOD training data. Also notice that despite in such unfavorable comparison, GNN-Safe largely exceeds Energy FT in the experiments of Table 2 and Fig. 2. This result serves as a strong evidence for the efficacy of our proposed model.

---

> > ### Author Response · Authors · 2022-11-10
> > **Response to Reviewer 2RDq (Part 2 of 2)**
> >
> >
> > > ***Q4: "One may train a binary classifier to classify the in-distribution samples and out-distribution samples into two different classes"***
> >
> > This idea is interesting yet orthogonal to our focus, given the above illustration. Again we focus on studying the reliability/safety of GNN node-level classifiers w.r.t. unseen OOD data for which it should be able to detect. If one additionally train a binary classifier to classify the ID and OOD data, it has no association with the GNN model for node classification, which is not what we are to study. Furthermore, additionally introducing an OOD classifier might increase the costs and in contrast, our work shows that the GNN classifier trained for node classification can still performs desirably for OOD detection, which is efficient and light-weighted. Therefore, the binary classifier is normally not conisdered in other peer OOD detection works [1-8].
> >
> >
> > > ***Q5: "The OOD exposure violates the principle of OOD detection because it becomes a supervised learning problem with OOD labels"***
> >
> > This comment is predicated on a hypothesis that our mainly focused setting assumes OOD exposure in training data, which is however a misunderstanding. Notice that we mainly focus on S1 which can be more difficult and closer to real applications (the OOD exposure is often hard to collect or define in practice.) Our main model GNN-Safe follows this setting and yields superior results in most cases (e.g., Table 2 and Fig.2). The improved variant GNN-Safe-r is to show how to allow further performance gain for the potentially existing S2 where extra OOD training data becomes available. The discussion for the later case is an extension and by-product of our work, not a major focus.
> >
> > One step back, even considering the OOD exposure, the problem is **not** "a supervised learning problem with OOD labels", since in our case, the (central) supervised task is for predicting node labels, and under the restriction of achieving desirable accuracy in the main task, our focused problem is how to enhance the reliability of GNNs against OOD data for which the model is expected to reject prediction.
> >
> > [1] Hendrycks et al., A baseline for detecting misclassified and out-of-distribution examples in neural networks, Arxiv 2016.
> >
> > [2] Liang et al., Enhancing the reliability of out-of-distribution image detection in neural networks, ICLR 2018.
> >
> > [3] Lee et al.,  A simple unified framework for detecting out-of-distribution samples and adversarial attacks, NeurIPS 2018
> >
> > [4] Hendrycks et al., Deep anomaly detection with outlier exposure, ICLR 2019.
> >
> > [5] Sun et al., React: Out-of-distribution detection with rectified activations, NeurIPS 2021.
> >
> > [6] Grathwohl et al., Your classifier is secretly an energy based model and you should treat it like one, ICLR 2020.
> >
> > [7] Zhao et al., Uncertainty aware semi-supervised learning on graph data, NeurIPS 2020
> >
> > [8] Stadler et al., Graph posterior network: Bayesian predictive uncertainty for node classification, NeurIPS 2021.

---

### Official Review · Reviewer_BViK · 2022-10-25

**Confidence:** 2
**Correctness:** 3
**Technical Novelty And Significance:** 3
**Empirical Novelty And Significance:** 3
**Recommendation:** 8

**Clarity, Quality, Novelty And Reproducibility:**

The motivation of the problem is clearly described in the introduction, and the proposed approach is easy to understand and well supported by several experiments and theoretical proof. The paper focuses on a novel energy-based belief propagation method for boosting OOD detection performance, which is meaningful. The experiment settings are detailed. However, there is no code in the supplementary materials.


**Strength And Weaknesses:**

**Strengths:**
1. The experiments in this paper are detailed in comparison with different methods under different datasets and GNN encoders.
2. This paper is well-written and easy to understand.
3. The authors provide proof for all theoretical results.

**Weaknesses:**
1. In the Introduction, it is said that “The scarcity of labeled training data in graph domain requires the model to exploit useful information from the unlabeled portion of observed data.” This statement is confusing because it is not the actual reason why semi-supervised learning is widely used in graph-based learning and performs well [1]. However, this paper claims to use semi-supervised learning to solve the problem of limited labeled data in graph datasets.
2. There is no detailed description of the evaluation metrics. In addition, GPN compares OOD-Acc for different methods, which is also an important metric to evaluate the performance of OOD data [2].
3. In figure 1, compared with chart (a), the distribution center of the green curve (in-distribution) is confusing in chart (b).
4. In Section 4.1, it is said that “use the subgraph DE as in-distribution data and other five subgraphs as OOD data for Twitch.” However, the authors don’t explain why different subgraphs follow different distributions.

[1] Graph-based Semi-supervised Learning: A review. Neurocomputing, 2020.
[2] Graph Posterior Network: Bayesian Predictive Uncertainty for Node Classification. NeurIPS, 2021.


**Summary Of The Paper:**

This paper proposed an intrinsic out-of-distribution (OOD) discriminator for semi-supervised learning on graphs based on an energy function called GNN-Safe, to enhance the reliability of the model against OOD data. The authors consider the interdependence between input data and other nodes in the graph rather than treating them independently. Experiments are compared with other models on five real-world datasets.

**Summary Of The Review:**

This paper is easy to read and proposes a novel approach based on the energy function to detect OOD data. Overall, this work is meaningful in OOD detection on graphs and can be further studied and optimized.

---

> ### Author Response · Authors · 2022-11-10
> **Response to Reviewer BViK**
>
> We are glad that you liked our paper and hope the following answers could help to improve your confidence.
>
> > ***Q1: "This statement is confusing because it is not the actual reason..."***
>
> Thank you for pointing out this confusion and suggesting the reference. We have modified this statement to "The scarcity of labeled training data in graph domain requires the model to exploit useful information from the observed structures that may reflect the underlying geometry behind data for better generalization [1]". The draft is modified accordingly.
>
> > ***Q2: "There is no detailed description of the evaluation metrics. In addition, GPN compares OOD-Acc for different methods."***
>
> We have added more descriptions for the evaluation metrics in Appendix B.3. The metrics AUROC, AUPR and FPR are commonly used for measuing OOD detection performance in the literature including our comparative models.
> As for the metric OOD-Acc additionally adopted by GPN, it is not suitable for our case. In the GPN work that focuses on uncertainty estimation, OOD-Acc (that refers to the testing accuracy on OOD data) is to evaluate if the model "can correct the anomalous features of OOD data" [2]. With a related yet different goal, the focus of our work is to enhance the reliability of GNNs: we mainly care about the testing accuracy on in-distribution data (the model is expected to give prediction) and detection for OOD data (the model should treat it as the outlier and reject prediction), following peer OOD detection works (MSP, ODIN, Mahalanobis, OE, GKDE, Energy).
>
> > ***Q3: "In figure 1, compared with chart (a), the distribution center of the green curve (in-distribution) is confusing in chart (b)"***
>
> This phenomenon is reasonable since in Arxiv dataset, both the in-distribution and OOD instance nodes are all inter-connected in a graph and the discrimination between them could be more difficult than Twitch where the ID and OOD nodes lie in different subgraphs. Another fact is that for Arxiv, the ID nodes spans from 1970 to 2015, which is a large time window, so the ID data could contain a mix of samples from more than one domain. Therefore, it is normal that the model's yielded energies exhibit two modes for ID data on Arxiv, which are overall lower than those of OOD data.
>
> > ***Q4: "However, the authors don’t explain why different subgraphs follow different distributions."***
>
> Thank you for raising this thoughtful question and we have added more explaination in Sec 4.1. We mainly follow the piorneering work [3] for creating OOD data in the experiments, and it has provided a detailed discussion regarding the different distributions of subgraphs in Twitch dataset in Appendix E.1 of their paper. One can refer to Table 4 of [3] to see that different subgraphs have distinct sizes, density and node degrees. Also, as shown by our exploratory experiments, the model trained on one subgraph (DE) performs undesirably on other subgraphs, which further verifies that their data distributions are different.
>
> Overall, we are encouraged that you appreciated the scope and potential impact of our work. We agree that OOD detection on inter-connected data is a significant research problem and as an initial exploration, our work could inspire more attempts in this new direction. We will release the full version of implementation upon publication to facilitate reproducibility.
>
> [1] Graph-based Semi-supervised Learning: A review. Neurocomputing, 2020.
>
> [2] Graph Posterior Network: Bayesian Predictive Uncertainty for Node Classification. NeurIPS, 2021.
>
> [3] Handling Distribution Shifts on Graphs: An Invariance Perspective, ICLR 2022.

---

> > ### Comment · Reviewer_BViK · 2022-11-26
> > **Response to the Authors**
> >
> > Dear authors,
> >
> > Thanks for your response.
> >
> > Thank you for adopting our comments regarding semi-supervised learning in the Introduction and adding a detailed description of the evaluation metrics.

---

### Official Review · Reviewer_QtVR · 2022-10-30

**Confidence:** 3
**Correctness:** 4
**Technical Novelty And Significance:** 3
**Empirical Novelty And Significance:** 3
**Recommendation:** 6

**Clarity, Quality, Novelty And Reproducibility:**

The paper is relatively clear and provides a number of significant advances over other approaches for OOD detection on graphs.

**Strength And Weaknesses:**

Strengths:
- The results seem significant over other state-of-the-art in OOD detection and generalization on graphs.
- Broad choice of GNN encoder backbones, competitor architectures, datasets and experimentation strategies. Multiple ablation studies included.
- Comparison of training times and inference times for all the models used included
- Each variant of their proposed models are compared with each other as well as competitors and a good summary/explanation of each of the experimental data is given.
- Have posited some insights on when regularization may be significant and when they may not help in OOD detection well citing examples from experiments.

Weaknesses:
- The gaps between methods that have access to OOD data to tune on and those that don't is significant. It would be helpful to better delineate these settings in the tables and results in the paper since the conditions in both cases is quite different.
- Provide further justification and rationale for the choice of methodology for creating synthetic OOD data, especially the label leave-out way in datasets like amazon-photos, cora and coauthor-cs.
- The explanation of how their approach is used to regularize learning in the supervised setting is not entirely clear.

​Typos
- Appendix B.1 in Co-author CS:
    - 6805 classes -> 6805 features


**Summary Of The Paper:**

The paper addresses an important problem of OOD detection and generalization for GNNs. An Energy-based discriminatory framework is explored that is shown to give a some significant advantages like model agnosticity, theoretical gaurantees and practically efficacy. This application and study of energy-based OOD detection methodology for inter-dependent data like graphs has not been done before and is novel.

They have started off with a simple instance-based energy function used to detect OOD samples, based on negative-log-likelihood loss. The authors show how OOD detection can be enhanced by propagating the energies calculated across neighbors for K-steps in a manner typical to message propagation in GNNs. Further, they have added a regularizing loss to this formulation that induces an energy boundary between in-distribution and OOD instances.

The experiments compare their variants with image-based and graph-based OOD competitors, with standard choice of metrics for detecting OOD and for calibrating accuracy. They show significant improvements in the OOD detection metrics without trading off the supervised learning of the models by too much. Their results improve upon previous work on OOD detection by a significant margin.


**Summary Of The Review:**

This paper provides a new approach for OOD detection and generalization on graphs that incorporates the graph's topology in an energy-based message passing scheme. The approach is well described and the rationale makes sense. The authors provide strong evidence that their approach improves generalization on a number of OOD tasks established on different graph benchmark datasets in both an unsupervised and semi-supervised setting.

---

> ### Author Response · Authors · 2022-11-10
> **Response to Reviewer QtVR**
>
> Thank you for the positive comments and constructive advice. We provide the answers below and modify the draft accordingly to increase your confidence.
>
> > ***Q1: "It would be helpful to better delineate these settings in the tables and results"***
>
> Thank you for the nice suggestion and highlighting the difference of the two settings. We have added a separate line in-between methods using or not using OOD exposure in Table 1 and 2.
>
> > ***Q2: "Provide further justification and rationale for the choice of methodology for creating synthetic OOD data"***
>
> The principle behind our adopted ways to create synthetic OOD data lies in the data-generative distributions of graph-structured data. Specifically, graph data contains three-fold information: node features X, graph structures A and node labels Y. So for OOD data construction, we study three types of distribution shifts regarding these three aspects, respectively. The "feature interpolation" embodies node feature shifts, the "structure manipulation" belongs to graph structure shifts, and the "label leave-out" targets node label shifts. Our consideration for using the "label leave-out" is that the OOD data generated in this way conforms to the true underlying data-generating process and introduces distribution shifts induced by different classes. Similar ways are also adopted by the literature, e.g., [1, 2].
>
> > ***Q3: "The explanation of how their approach is used to regularize learning in the supervised setting is not entirely clear"***
>
> Thank you for mentioning this vagueness which may help us to improve the presentation. For our energy-regularized learning case, the data used for training contains two portions: one is in-distribution (ID) data and another is out-of-distribution (OOD) data. The model GNN-Safe-r is trained with a mix of two losses $L_{sup}$ and $L_{reg}$. $L_{sup}$ is a standard MLE loss (defined by Eqn. 6) on the ID training data, the same as standard GNN models. $L_{reg}$ is the introduced regularization loss (defined by Eqn. 9) that is computed by the estimated energy of our GNN model on both the ID and OOD training data. Please let us know if there is any further question on this point.
>
> We modify the typo in the revision and thank you for pointing this out.
>
> [1] Uncertainty Aware Semi-Supervised Learning on Graph Data, NeurIPS 2020.
>
> [2] Graph Posterior Network: Bayesian Predictive Uncertainty for Node Classiﬁcation, NeurIPS 2021.

---

### Public Comment · ~Benedek_Andras_Rozemberczki1 · 2022-11-05
**Misattribution of Twitch Gamers**

The Twitch-Gamers dataset was proposed in this paper:

```bibtex

>@misc{rozemberczki2021twitch,
       title = {Twitch Gamers: a Dataset for Evaluating Proximity Preserving and Structural Role-based Node Embeddings},
       author = {Benedek Rozemberczki and Rik Sarkar},
       year = {2021},
       eprint = {2101.03091},
       archivePrefix = {arXiv},
       primaryClass = {cs.SI}
       }

```

---

> ### Author Response · Authors · 2022-11-06
> **Thank you for providing the reference**
>
> We will fix the citation for the used Twitch dataset during the revision. Thanks for letting us know.

---

### Author Response · Authors · 2022-11-10
**General Response by Authors**

Dear Area Chairs and Reviewers,

We appreciate the reviewers' time, valuable comments and constructive suggestions. Overall, the reviewers recognized our work clearly written (QtVR, BViK, 2RDq, PiYE) and well motivated (QtVR, BViK, 2RDq), and appreciated our reasonable methods (QtVR, BViK, PiYE) and strong empirical results with extensive experiments (QtVR, BViK, 2RDq, PiYE).

However, we noticed that the reviewers currently tend to have divergent opinions on our novelty. Reviewer QtVR and BViK thought we explore an "important problem" that "has not been done before" with "novel" methods and is "meaningful" to be "further studied and optimized" by future works. Still, Reviewer 2RDq and PiYE have reservation on our contributions and differences with existing works, based on which we believe that there might exist some potential misunderstandings that affect how our work is interpreted.

We next provide detailed answers to all the specific questions raised by the reviewers. Further discussions are welcomed to facilitate the reviewing process towards a comprehensive evaluation of our work.

---

### Decision · Program_Chairs · 2023-01-20

**Decision:**

Accept: poster

**Justification For Why Not Higher Score:**

NA

**Justification For Why Not Lower Score:**

NA

**Metareview: Summary, Strengths And Weaknesses:**

**Summary** This paper studies out-of-distribution detection in the context of graph neural networks (GNNs). This direction of OOD detection with graph-structured data is important, and can be of interest to both graph learning and OOD detection communities. Hence, I am very much in support of this paper's motivation and positioning.

The core idea of this paper is a nice extension of Liu et al. 2020 [1] from CNNs to GNNs, supported by extensive evaluations and promising results. The methodological novelty was questioned by reviewer 2RDq. During the rebuttal, the authors provided further clarification on the novelty. Compared to [1], the proposed method is designed to handle inter-dependent rather than i.i.d. data, and further, the paper introduces energy belief propagation (which seems new in the context of GNNs). The AC recommends the authors revise the novelty statement to highlight these two points more clearly.

----------------------
**Suggestions for revision [important]** It seems to me the paper draws heavily on the intellectual idea in [1] (both the energy score and energy-based regularization loss). This is fine but they probably should give better credit to the source of inspiration. Because of the lack of a clear attribution of intellectual merit, several places can be flagged as potential plagiarism. I provide a few concrete examples below:

* Page 5, Equation (9), the energy-regularized loss function is mathematically identical to the square hinge loss in [1], though the paper did not cite the paper where the loss originates. Notion wise, the slight change was from $\max(0, x)$ to $\text{ReLU}(x)$. The loss function should not be framed as a new contribution in this paper, since Section 3.3 in spirit follows largely Section 3.3 in [1].

* Page 12, the proof technique of Proposition 1 is identical to Page 4 in [1], which showed the negative log-likelihood loss pushes down energy for in-distribution data. This should be cited properly.

* Page 4, Equation (5), _"...More importantly, without changing the parameterization of the graph neural network,
we can express the free energy function $E(\mathbf{x})$ in terms of the denominator of the softmax activation"_. Note that the sentence is a word-by-word copy of the same sentence on Page 3 of [1], except for replacing "neural network" with graph neural network.

With that being said, I still think the merit of the paper outweighs the ethical concerns in writing (which are fixable). I thus invite the authors to carefully revise the final version, incorporating the points raised above.

------------------------------
**Recommendation**
Three reviewers recommend acceptance while one reviewer rated the paper marginally below the acceptance threshold. Overall, the AC believes the paper provides valuable contributions to the community. I would like to recommend acceptance, conditioned on the suggested changes incorporated in its final version.

[1] Liu et al. Energy-based Out-of-distribution Detection, NeurIPS 2020.




**Note From Pc:**

if the above contains the word "oral" or "spotlight" please see: "oral" presentation means -> notable-top-5% and "spotlight" means -> notable-top-25%. As stated in our emails, we are disassociating presentation type from AC recommendations